# Effects of Traditional and Modern Post-Harvest Withering Processes on the Composition of the *Vitis v.* Corvina Grape and the Sensory Profile of Amarone Wines

**DOI:** 10.3390/molecules26175198

**Published:** 2021-08-27

**Authors:** Diego Tomasi, Andrea Lonardi, Davide Boscaro, Tiziana Nardi, Christine Mayr Marangon, Mirko De Rosso, Riccardo Flamini, Lorenzo Lovat, Giovanni Mian

**Affiliations:** 1Council for Agricultural Research and Economics—Research Centre for Viticulture and Enology, CREA-VE Viale XXVIII Aprile 26, 31015 Conegliano, Italy; diego.tomasi@crea.gov.it (D.T.); davide.boscaro@crea.gov.it (D.B.); tiziana.nardi@crea.gov.it (T.N.); mirko.derosso@crea.gov.it (M.D.R.); riccardo.flamini@crea.gov.it (R.F.); lorenzo.lovat@crea.gov.it (L.L.); 2Bertani Domains Società Agricola A R.L., Via Asiago 1, 37023 Grezzana, Italy; andrea.lonardi@bertani.net; 3Department of Agronomy, Food, Natural Resources, Animals and Environment (DAFNAE), University of Padova, Viale dell’Università 16, 35020 Padova, Italy; christine.marangon@unipd.it; 4Department of AgriFood, Environmental and Animal Sciences, University of Udine, Via delle Scienze 206, 33100 Udine, Italy

**Keywords:** post-harvest, grape, wine, withering, stilbenes, aroma, Amarone, Corvina

## Abstract

In the Valpolicella area (Verona, Italy) *Vitis vinifera* cv. Corvina is the main grape variety used to produce Amarone wine. Before starting the winemaking process, the Corvina grapes are stored in a withering (i.e., dehydrating) warehouse until about 30% of the berry weight is lost (WL). This practice is performed to concentrate the metabolites in the berry and enrich the Amarone wine in aroma and antioxidant compounds. In compliance with the guidelines and strict Amarone protocol set by the Consorzio of Amarone Valpolicella, withering must be carried out by setting the grapes in a suitable environment, either under controlled relative air humidity (RH) conditions and wind speed (WS)—no temperature modification is to be applied—or, following the traditional methods, in non-controlled environmental conditions. In general, the two processes have different dehydration kinetics due to the different conditions in terms of temperature, RH, and WS, which affect the accumulation of sugars and organic acids and the biosynthesis of secondary metabolites such as stilbenes and glycoside aroma precursors. For this study, the two grape-withering processes were carried out under controlled (C) and non-controlled (NC) conditions, and the final compositions of the Corvina dried grapes were compared also to evaluate the effects on the organoleptic characteristics of Amarone wine. The findings highlighted differences between the two processes mainly in terms of the secondary metabolites of the dried grapes, which affect the organoleptic characteristics of Amarone wine. Indeed, by the sensory evaluation, wines produced by adopting the NC process were found more harmonious, elegant, and balanced. Finally, we can state how using a traditional system, grapes were characterised by higher levels of VOCs (volatile compounds), whilst wines had a higher and appreciable complexity and finesse.

## 1. Introduction

Wine grapes are generally harvested at the proper technical ripening stage, which is when the major chemical ripening parameters (pH, acidity, soluble sugars, polyphenols and aromatic compounds) meet the oenological requirements of each type of wine, according to the oenological goal set by the winemakers and their focus on the market where they are selling [1]. Indeed, some specific wines (sweet or dry) are produced by drying the harvested grapes in the open air (Moscato di Pantelleria) or by storing them in withering (dehydration) chambers before vinification, to achieve a weight loss (WL) of around 30% (Pedro Ximenez of Montilla-Moriles, Riesling, Commandaria of Cipro, Vin Santo di Santorini, Valpolicella Amarone).

Worldwide, Amarone is considered one of the most important Italian wines (15 million 750 mL bottles, with an average commercial price of 62 USD/bottle—Consortium for the protection of DOP Valpolicella Wine, 2019 data provided by Siquria). Amarone is produced from three native varieties—Corvina, Corvinone, and Rondinella (Consorzio Valpolicella, 2010); however, withered Corvina grapes are still the main variety used to craft this wine [2]. Corvina is a native variety of the Verona area; the wine it gives is characterised by hints of cherry, bitter almond, blossom aromas, and refreshing acidity. Nevertheless, it lacks anthocyanins and has a moderate content of skin tannins. The withering process, in this sense, has been selected over the decades to concentrate and enrich the berry and wine composition.

The grape drying process involves two simultaneous transfer phenomena: (i) heat (energy) transferred from the environment to the bunches and (ii) water (mass) transfer from the inside to the outer surface of the berries, followed by evaporation. The drying conditions influence the properties of the resulting grapes and wines; slow drying at a low temperature (T) and aeration rate (W) and higher relative humidity levels (RH) provides more harmonious wines (e.g., equilibrated, balanced ratio of every aromatic and chemical compound) [3], whereas fast drying (high T, low RH, high W), could lead to wines with unbalanced aromatic profiles [4]. The fact of needing to choose one or another dehydrating system is an instance for what winegrowers in Amarone wine production area struggle, due to the fact that for some of them a fast drying may be important in order to speed up each winery operation but, at the same time, might result in a lower wine quality. Drying alters the overall structure of the berry due to the loss of moisture and, consequently, the berry sanitary status is also affected [5]. As a consequence of berry water loss, the skin/pulp ratio increases; moreover, withering causes compositional changes in the phenolic and aromatic composition to an extent that is variable depending on the specific drying conditions [6]. Biochemical changes have also been described in berries dehydrated in ventilated chambers, deriving from environmental parameters and endogenous factors (genotype) [7].

The most marked effect occurs on sugars. Their increase is mainly due to a concentration effect; nevertheless, the composition and nature of the sugars change throughout the drying process, particularly in terms of the glucose/fructose ratio variations [8]. The change in the glucose/fructose ratio in this phase is mainly due to respiration [9], which preferably uses glucose as a substrate. Organic acidity, as for sugars, during the drying process is influenced by the concentrative aspect, and both malic and tartaric acid are affected by the metabolic respiration and precipitation processes. These changes may have a different intensity depending on the organic acid considered and the stage and intensity within the dehydration process. It is reported that there are slight increases in the total acidity mainly due to the concentration of tartaric acid [10] as well as a slight decrease in malic acid [11]. Specifically, during the drying process, the sugar/acid ratio varies significantly, and this is linked to the marked concentration of sugars [12].

Alongside the primary metabolism compounds, the secondary metabolites, mainly present in the hypodermal layer of the berry skin and that affect pigmentation, flavour and aroma of the wine, also need to be considered [13,14]. The withering process affects the development of dehydration-related aromas and polyphenolic compounds [6]. Following harvest, as the grape berry is metabolically active until cell death, it reacts to endogenous and exogenous stresses such as dehydration. In the early stages of the drying process, there is an enrichment of polyphenols [15,16,17] due to de novo biosynthesis confirmed by the increase in the transcripts of their biosynthetic pathway [18]. At a later stage, polyphenol oxidation takes place, leading to a depletion of many of them [16,19]. Additionally, while withering does induce a general decrease in polyphenols, such as anthocyanins, flavanols, procyanidins and glycoside flavonols [19,20], an increment of other polyphenols, such as trans-resveratrol (3,5,4′-trihydroxystilbene), taxifolin, quercetin, some methoxylated flavanones and acylated anthocyanins, was observed [9,19,20,21]. Finally, the synthesis of stilbenes is a response to abiotic cell stress (i.e., *trans*-resveratrol, viniferins, *cis*- and *trans*-piceid) [15].

Likewise, piceatannol (3,4,3′,5′-tetrahydroxy-*trans*-stilbene) has been shown to block LMP2A, a viral protein-tyrosine kinase associated with leukaemia, non-Hodgkin’s lymphoma and other diseases associated with the EBV virus, which also acts on human melanoma cells [22]. The withering process also affects the volatile organic compounds (VOCs) and their glycoside precursors in grapes [23,24]. Indeed, following harvest, the dehydrated berry VOCs increase not only as a concentration effect but also as a consequence of an active process [25]. It was observed that a dehydration temperature of above 30 °C drastically reduces the share of primary and varietal aroma compounds (e.g., terpenes and norisoprenoids), which instead are maintained and concentrated slowly at temperatures of around 20 °C and up to 40% of berry water loss [26]. Studies on aroma composition carried out during the dehydration of Pinot noir [27] and Pedro Ximenez [28] grapes showed that the aroma profiles are affected both qualitatively and quantitatively due to the development of new odorous compounds [23]. These new compounds confer notes of jam, raisins, plums, morello cherries, and almond. The main grape aroma compounds of Amarone wines belong to chemical classes of C_13_-norisoprenoids, terpenes, benzenoids, furans and aliphatic alcohols [29].

Current Amarone withering techniques require an average of 2.5 months (as stated by law, grape crushing cannot be done before 1st December, MIIPAF; Reg, n° 558533/2019) [30], yielding a final product that is richer in sugars, polyphenols, and aromatic compounds. In the past, the Amarone withering was carried out by setting the grapes on particular wooden trays known as “arelle” inside dehydration chambers not provided with any climatic control; nowadays, air humidity and ventilation can easily be controlled by using dehumidification systems along with fans for air circulation (no extra temperature conditioning is allowed by the Amarone production regulations) [30]. Indeed, artificial environment conditioning provides faster withering and allows healthy dried grapes to be achieved, although this seems to be stressful for the berries [7]. Studies focusing on post-harvest events have confirmed that functional biochemical and molecular changes continue within the berries reflecting the environmental conditions of post-harvest ripening and/or senescence [18,31,32]. Moreover, Ferrarini [33] conducted a study to investigate the best timing for harvest and withering to achieve a more aromatic Amarone wine.

Given these circumstances, the choice between a controlled (modern) or non-controlled (traditional) system should be pondered by winemakers, since it can hardly influence the berry composition and final wine profile. Yet the withering method and its effects on the above-mentioned berries and wines composition is not well studied and need to be clarified in order to give precise information to winemakers. However, these issues will also be discussed by local authorities. Hence, the primary purpose of this work is to compare natural and artificial Corvina withering processes to assess their effect on dried grape chemical composition and peculiar wine organoleptic evaluations. This aspect is particularly interesting for the few wineries that still adopt the traditional drying synthesis to pursue a true-to-varietal character and vintage-related wine.

## 2. Materials and Methods

### 2.1. Experimental Setup and the Two Whithering Rooms

The experimental data were collected during 2016 and 2017. The vineyard was selected in the Valpolicella classic area (Novare locality in the Negrar municipality, Verona. 45.535468 North (N), 10.943691 East (E)), Bertani Winery. The location climate is typical of a temperate area located in the Northeast of Italy, with warm summers and cool winters. Annual rainfall is approximately 850 mm, concentrated mainly in spring and autumn. As for temperatures, during the growing season (April–September), the average is about 20–21 °C. Indeed, concerning the vineyard, the vine training system adopted was Guyot with a plant density of 4444 vines per hectare; the variety raised was Corvina, clone ISV-CV48 grafted onto Kober 5 bb, grown predominantly in calcareous soil. Corvina was chosen as it is commonly the main variety for Amarone wine production. Harvest, following local tradition, was scheduled to take place ten days before the standard harvest period for Valpolicella fresh wine, occurring on 17 September in 2016 and on 6 September in 2017. In order to preserve their healthy status, the carefully selected grapes were placed in plastic trays—in the case of the modern warehouse drying process—and in the traditional “arelle” (wooden trays) to wither the grapes under natural conditions. The two withering processes were carried out in two different rooms inside the winery:

(A) For the modern warehouses (C process), the appropriate use of an air dehumidifier machine (GOBI Cornwall Electronics 50L/GG, Cornwall Electronics Ltd., Valecia, Spain) maintained the air humidity inside the room between 60% and 80% (optimal range for grape dehydration, as described by Amarone disciplinary [30]), and fans (Extractor Fan TT Smart, Cornwall Electronics Ltd., Valecia, Spain) were used to provide airflow through the drying trays. No temperature control was applied. 

(B) Instead, a traditional dehydration process (NC) took place without any air humidity control system or mechanical airflow: the warehouse supervisor was in charge of opening (during the night and dry days, i.e., RH and T over the chosen range inside chambers) or closing (rainy or humid days, i.e., when the external RH was over the range) the windows, to enable the high-moisture air to escape and the dry, fresh air to flow inside the drying chamber (Appendix A). In the experimental conditions, the ratio between room volume/grape quantity was the same in the two warerooms. 

Berry chemical composition was analysed at four different stages: (i) T0 fresh berry, (ii) −10% of berry water loss (WL), (iii) −20% WL and (iv) −30% WL, as described in Section 2.3 below.

### 2.2. Withering Environmental Conditions

The climatic conditions in both environments, C and NC, were measured. Specifically, temperature and humidity were measured using air sensors (Lascar Electronics EL–USB–2, Bergamo (BG), Italy) located in each environment of the study: four sensors per drying chamber, two in contact with the withering grapes and two 50 cm above the grapes. The daily values recorded were the average of the measurements collected every 15 min. The outdoor climatic conditions were measured by the weather station belonging to Avepa (Avepa.it) located in the Negrar area. In 2016, due to contingent technical problems during data download, the initial climatic records (first ten days) collected in the withering chamber were lost.

### 2.3. Kinetics of Withering and Grape Samples

The kinetics of the withering process was evaluated by measuring the post-harvest weight loss; in the NC and C treatments, values are expressed as the percentage (%) of weight loss during withering. Specifically, data were taken by weighing two arella trays of about 250 kg of grapes in the case of the natural environment treatment, and two other pallets of trays (around 200 kg of grapes) with regard to the controlled environment. The arella trays and the pallets were chosen from two representative positions within the drying chambers. As a final consideration, we report that these values do not come from replicates, since they represent the whole weight of a tray (with hundred bunches), only distinguished in C and NC.

For both drying processes (C and NC), at harvest time (T0) and during withering (−10% WL, −20% WL, and −30% WL), three samples of grapes weighing 1.0 kg each were collected to determine the berry chemical composition. The samples were obtained from several portions of the grape bunches stored on the selected trays and pellets. Samples were taken every 4 days, thus enabling us to understand when the different WL stages were reached.

### 2.4. Quantification of Sugars and Acids

The total soluble sugars in the berries were quantified using an ATAGO PR-32 digital refractometer (Fischer Scientific, Milano (MI), Italy) (0–32%) and expressed in Total Soluble Sugars (TSS), Brix degree; in the same samples, the organic acid profile of the berries (specifically tartaric and malic acids, expressed in g/L) was determined by high-pressure liquid chromatography (HPLC Agilent 1220 infinity, ThermoFischer Scientific, Abingdon, UK). The samples for HPLC were prepared by taking 250 µL of grape must diluted 1:50 with distilled water. The sample was then filtered through a 0.2 μm filter and analysed. The grape must samples were prepared by pressing three subsamples of 250 g of berries. 

### 2.5. Quantification of Pigments

Following the method described by Di Stefano [11], in 30-berry samples per withering process, the total flavonoids and total anthocyanins were quantified. In fact, among the berry skin flavonoids, those belonging to the anthocyanins class are the most interesting in terms of direct feedback on the characteristics of a wine. The values were expressed in mg/kg of grapes.

The aglycones liberated from glycoside-bound aroma precursors were analysed by gas chromatography/mass spectrometry (GC/MS, EI 70 eV, ThermoFischer Scientific, UK) after performing enzymatic hydrolysis as previously described [34,35,36]. The skins of 100 berries were separated from the pulp and extracted with 35 mL of methanol for 4 h in the dark. The extract was homogenised using an Ultra-Turrax (IKA^®^-Werke GmbH & Co. KG, Staufen, Germany) and centrifuged. The volume of the supernatant was adjusted to 250 mL by addition of distilled water, and the solution was treated with 2 g of insoluble poly(vinylpyrrolidone) (PVP) to reduce the polyphenolic content. Additionally, the pulp was homogenised and centrifuged, and the volume was adjusted to 250 mL. The solutions were treated with 75 mg of pectolytic enzyme for 4 h at room temperature to release the volatile compounds. A volume of 200 μL 1-heptanol 250 mg/L solution was added to the extract as internal standard, and the aglycones were isolated by solid-phase extraction using a 10 g C_18_ cartridge (Waters Corporation, Milford, MA, USA) previously activated by successive passages of 30 mL dichloromethane, 30 mL methanol, and 30 mL water. Salts, sugars, and other polar compounds were removed by washing the cartridge with 50 mL of water, and the fraction containing the free compounds was recovered using 50 mL of dichloromethane. The solution was concentrated to 2–3 mL by distillation using a 40 cm length Vigreux column then to 200 μL under a nitrogen flow prior to performing GC/MS analysis.

Gas chromatography/mass spectrometry (GC/MS) analysis was performed using a 6850-gas chromatography system by Agilent Technologies (Santa Clara, CA, USA), fitted with a fused silica HP-INNOWax polyethylene glycol capillary column (30 m × 0.25 mm, 0.25 μm inner diameter) (Agilent Technologies, Santa Clara, CA, USA), coupled with HP 5975C mass spectrometer and 7693A automatic liquid sampler injector (Agilent Technologies, Santa Clara, CA, USA). Compound identification was performed using the NIST Mass Spectral Libraries Database (rev08) and the in-house CREA-VE database. To perform a comparison among the samples the contents of volatile compounds were expressed as μg of internal standard per kg of dried grape (d.g.).

### 2.6. Analysis of Stilbene Compounds

For our study, only visually healthy grapes, including berries affected by little or no stilbene-oxidase activity caused by (e.g.,) botrytis infections were collected. Consequently, differences observed in the two drying processes were mainly related to the different dehydration conditions. Sample preparation for High-Pressure Liquid Chromatography—diode array detector (HPLC/DAD) analysis was performed using the methods already described but modified to our goals [37,38]. The skins of 20 berries were homogenised with 45 mL of methanol using a T25 Ultra-Turrax^®^ and kept under stirring for 20 min in the dark at room temperature. The extract was centrifuged, and to the supernatant of 200 μL of trans-4-hydroxystilbene, a 9.6 mg/L solution in methanol was added as internal standard. The solution was evaporated to dryness under vacuum at 40 °C, and the residue was suspended in 10 mL of HCl 10^−3^ M in water. After the addition of 5 g of NaCl, the solution was extracted to 3 × 5 mL with ethyl acetate. Ethyl acetate extract was evaporated to dryness under vacuum at 40 °C and reconstituted with 4 mL of methanol/aqueous H_3_PO_4_ 5 × 10^−3^ M 30:70 (*v*/*v*) solution. Finally, the sample was filtered through a 0.22 μm PTFE filter and analysed by HPLC using a 1220 Infinity G4290B pump coupled with a Gilson 170 G1315A DAD-UV detector (Agilent Technologies, Santa Clara, CA, USA) equipped with an RP C18 column (ODS Hypersil^®^ 200 mm × 2.1 mm i.d., 5 μm, Thermo Hypersil-Keystone, Bellefonte, PA, USA). Elution was performed using a binary solvent composed of A) methanol and B) H_3_PO_4_ 5 × 10^−3^ M in water and the following gradient program: from 25% to 30% of A in 30 min, from 30% to 35% of A in 20 min, from 35% to 75% of A in 10 min, from 75% to 85% of A in 15, from 85% to 25% of A in 5 min, and isocratic for 20 min (flow rate 0.25 mL/min, injection volume 10 L). *trans*-Piceid, *trans*-resveratrol, *trans*-piceatannol, *trans*-εε-viniferin and δ-viniferin were quantified by recording the chromatograms at 307 nm, *cis*-piceid at 285 nm. The UV–Vis spectra in the 200–400 nm range were also recorded. Standard of *trans*-resveratrol, piceatannol, *trans*-piceid and *trans*-4-hydroxystilbene were purchased from Sigma–Aldrich (Milan, Italy); δ-viniferin was provided by CT Chrom (Marly, Switzerland). *cis*-Piceid was produced by photoisomerisation of trans isomer; *trans*-ε-viniferin was extracted from a lignified vine cane of Gamaret following the method described by Pezet et al. [39].

Qualitative profiles of stilbenes were characterised by using an Ultra-High Performance Liquid Chromatography (UHPLC) Agilent 1290 Infinity system coupled to Agilent 1290 Infinity Autosampler (G4226A) and Agilent 6540 accurate-mass Quadrupole-Time of Flight (QTOF) Mass Spectrometer (nominal resolution 40.000) equipped with Dual Agilent Jet Stream Ionization source (Agilent Technologies, Santa Clara, CA, USA). Analyses were performed by using the methods and analytical conditions previously described [40,41].

### 2.7. Winemaking and Wine Tastings

The winemaking was made via micro-vinification carried out at the vinification centre of the cooperative nurseries of Rauscedo (VCR), where about 150/180 kg of dried grapes were used for each vinification obtaining about 60 L of wine, as previously described [42]. The vinification process started with the collection of withered grapes when the 30% weight loss was reached, after which the clusters were crushed and pressed to obtain the must. The must was put in a fermenter with the addition of 10 g/hl of potassium metabisulfite, 0.3 g/t of enzymes (Lysis first^®^, Oenofrance, Montebello Vicentino, Italy), 5 g/hl of ascorbic acid and 0.5 g/t of tannin. Afterwards, a sample of must was picked up and inoculated with 20 g/hl of selected yeast (Zymaflore FX10^®^ and F83^®^, Laffort, Paso Robles, CA, USA), with the addition of 2 g/t of thiamine and vitamins (Oenofrance), 2 g/t of yeast extracts rich in amino acids (Oenofrance) and 2 g/t of inactivated yeast and cellulose (Oenofrance) and then re-mixed incorporating it in the fermenter. After 48 h from the inoculation, to favour malolactic fermentation, 1 g/hl of *Oenococcus oeni* bacteria (Lalvin VP41^®^, Lallemand, Montreal, QC, Canada) was added. Once a determinate alcoholic level had been reached in the must (12/13 alcohol degrees), a high-alcohol-tolerant yeast at a concentration of 30 g/hl was added (Lalvin 2226^®^, Lallemand), also adding 15 g/hl of mineral nutrients (Vivactiv Performance^®^, Oenofrance). The use of this second strain of yeast is necessary to reinforce the fermentation under high alcoholic level. At the end of the fermentation, a soft pressing of grapes was performed and 5 g/hl of tannin (Tannino Perfect^®^, Oenofrance), 5 g/hl of potassium metabisulfite, 15 g/hl of yeast hulls (Vivactiv Control^®^, Oenofrance) and 10 g/hl of nutrients (Philya LF^®^, Oenofrance) was added. After having decanted the wine for 3 days, the malolactic fermentation was favoured bringing the must at 21 °C. At the end of the malolactic fermentation, another decanting was performed, and 2, 5 g/hl of enzyme (Lallzyme MMX™, Lallemand) was added. The temperature was reduced to 10 °C, and for four weeks a batonnage was performed twice per week. At the end of the batonnage, another decanting operation was performed, and the same temperature was kept for two weeks. At the end of this period, the wine was filtered with cardboard filters, stabilised with a stabiliser (Cryokappa^®^, Oenofrance), and brought to −5 °C for 3 weeks. After that, the wine was filtered again with a, firstly, 1 µm, and secondly, 0.45 µm, candle filter and bottled.

The tasting analysis was carried out in Conegliano (TV) at CREA-VE by a test panel made up of 10 specialist judges trained properly plus a panel leader. To test and to confirm the reliability and accordance of the judges, a periodic training among the participants to the panel test was made following precise criteria reported in the literature [43,44]. For the quantitative evaluation of the intensity of attributes (olfactory, gustatory-tactile and retro-olfactory), Quantitative Descriptive Analysis was used [45] with the help of a question sheet (Appendix A), providing discrete scale responses with intervals from 1 to 9. For each of the three attributes, the relative differences between wines were analysed and confirmed, submitting the judgments to statistical analysis using the ANOVA method (F-test *p*-values < 0.0001 [46]). The 2016 and 2017 wines were tasted one year following vinification; the wines were stored in stainless steel tanks in a temperature-conditioned cellar at 12–14 °C.

### 2.8. Statistical Analysis

One-way analysis of variance was performed using “R” freeware. Statistical analysis to determine significant differences between treatment means was carried out using the Student-Newman-Keuls test (*p* ≤ 0.05).

## 3. Results

### 3.1. Temperature–Humidity Mean and Kinetics of Withering (WL) in Two Years of Study in C and NC Environments

In order to describe the kinetics of withering, namely the velocity of WL and sugar concentration, using climate data collected inside and outside the withering chambers, mean temperature and mean humidity per day were measured throughout the two test years. The weather conditions of the test locality (Negrar) were also taken into account to understand the difference between treatments inside and outside the chambers. In 2016 (Figure 1), the C chamber’s average temperature values were higher than in the NC one. On average, the temperatures in the chamber conducted in NC were closest to the outside ones. The relative humidity was comparable in the two conditions C and NC, though C presented a slightly lower level than NC, mainly at the beginning of the process when the WL from the berries was higher, while the external treatment followed the normal trend of the daily weather conditions.

As for 2017 (Figure 2), the average temperature was again higher in C and lower in NC, and in this case, NC was also closest to the external one. As air humidity was controlled in the C withering chamber, this parameter was lower in C conditions and higher in NC conditions, while the external humidity level presented more abrupt variability.

As previously reported, T and RH are fundamental for the kinetics of WL. Hence, measurements of the kinetics of withering were taken and are indicated as a percentage of weight loss during the period of withering, starting from time 0 (grape set-aside) to 30% WL (Figure 3). 2016 evidenced a faster withering in C than NC throughout the entire period, and −30% WL was reached ±17 days earlier in C than in NC, due to the constant differences in T and RH. In 2017, there were no significant differences between treatments during the first period of withering (−20% WL), since the T and RH were similar in the first period, but at the end, −30% WL was once again reached earlier in C (±14 days) than in NC.

### 3.2. Sugar Accumulation Trend

During withering, generally, a concentration of sugars occurs due to the berry water loss. The analysis of the two tested years reported a similar trend (Figure 4). In 2016 the grape set-aside started on 17 September; in this year, NC grapes showed a slower accumulation of sugars than C, but at the end (−30% WL), there were no differences in sugar concentration between treatments. In 2017, the grape set-aside started on 6 September, and the trend of sugar enrichment was similar. In this year, in NC there were more degrees than in C, during all the periods. The berry sugar content was 30.2 and 30.8 Brix degree in 2016, 27.0 and 28.6 in 2017, for the C and NC conditions, respectively. At −30% WL, the NC treatment showed a higher degree than C, explicitly highlighted in 2017. In 2016, the outside RH was higher than in 2017; as a consequence, C conditions showed markedly lower room RH compared to NC, which can explain the fact that C sugar accumulation was faster in C conditions. In 2017, the external weather conditions were different from those in 2016. Indeed, the air RH was much lower, and it was easier to control the room RH even in the NC treatment (see the similar level in RH between C and Nc reported in Figure 2), which can explain the similar trend in sugar accumulation. The last two considerations refer to the final sugar enrichment, which tends to be higher in NC conditions, and to the fact that the first stage of sugar enrichment tends to be more active in C conditions.

### 3.3. Accumulation of Acid Trend and Content in Specific Acid Compounds

Because of water evaporation and WL, acids tended to accumulate during withering. The analysis of the two tested years reported a similar accumulation trend in terms of total acidity (Figure 5). In 2016, the trend was similar in both treatments, with only a minor decrease in NC at the beginning of the process, but the acidity in NC was slightly above C with no significant differences during the final stage of the process. The trend was confirmed in 2017, where acidity levels were very close during the whole period of the drying process, except for the last stage when the NC environment displayed higher acidy with a final significant difference. Moreover, regarding the components of acidity (Figure 6), in 2016, the level of tartaric acid in NC remained the same, but during the final stage (−30% WL), it increased, while in C, there were no significant differences throughout the WL process. As for malic acid, there were no differences between C and NC at the different stages of WL. In 2017, at the different stages of WL, tartaric acid tent to be consistently higher in NC than C, even though no statistical significance was found. The highest content was found at −30% WL for NC. Regarding malic acid, the concentration was higher in the second part of the drying process, with no difference between the two treatments.

### 3.4. Pigment Accumulation Dynamics

Different climate conditions inside the withering chambers may affect the berry pigment compounds. At the end of the process, the content in total polyphenols consistently predominated in NC in the two tested years of study (Figure 7 and Figure 8). In both years, total polyphenols were higher in NC at −30% WL (2586 mg/kg berries t.q. in C vs. 2933 in NC in 2016, 1550 in C vs. 2.000 in NC in 2017). Specifically, in 2016, the accumulation trend was similar in C and NC; furthermore, the differences between the two treatments were significant only at −30% WL (*p* ≤ 0.05). In 2017, the trend was again similar in C and NC, but the differences became highly significant starting from the 10% WL stage. As for anthocyanins concentration, in 2016, there were no differences during the initial stage (−10% WL), whilst at the end of the drying period, NC grapes had 11% more anthocyanins than C grapes (614 mg/kg berries t.q. and 549 for C and NC, respectively). In the second year (2017), there was a considerably higher and more meaningful content in NC than C at every stage of WL (at −30% WL: 242 mg/kg berries t.q. in C vs. 380 mg/kg in NC, +36%).

The anthocyanin profile of Corvina is constituted by malvidin, peonidin, petunidin, delphinidin and cyanidin monoglucosides and their acyl derivatives, where malvidin-3-*O*-monoglucoside is the main compound. No relevant differences between the two processes in the qualitative profile of anthocyanins were observed (data not shown).

### 3.5. Synthesis of Stilbenes

Withering induces specific chemical changes in the berry, and most of them involve secondary metabolites. In particular, the biosynthesis of stilbenes is promoted by abiotic stress—e.g., cell dehydration—and these compounds determine the nutraceutical properties of the grapes and wines. The profile of stilbenes in Corvina grape before withering was characterised by liquid chromatography/quadrupole-time of flight mass spectrometry (UHPLC/QTOF). Nineteen compounds were identified; they include *trans*-resveratrol, *cis* and *trans*-piceid, piceatannol, *E* and *Z* astringin, pallidol, *E* and Z ε-viniferin, *Z*-ω-viniferin, δ-viniferin, a resveratrol dimer, caraphenol, pallidol-glucoside, α-viniferin, *E* and *Z* miyabenol C and two resveratrol-tetramers (Appendix A). Stilbene derivatives for which the standards were available were quantified at the different withering stages by performing HPLC/DAD analysis. The contents of *trans*-resveratrol and total stilbenes during the two processes in the two harvests are shown in Figure 9.

In 2016, a constant increase in *trans*-resveratrol and total stilbenes were observed throughout both withering processes. In particular, until the −20% WL stage, stilbenes were higher in the NC samples. In the last stage (between 20–30% WL), the level of stilbenes in NC samples remained constant, while in the C samples, it continued to increase, reaching values comparable to those in the other treatment. In the latter year, the trend was different: at −10% WL, both processes induced a low stilbene increase, and the levels remained similar until the −20% WL stage. A sharp increase in the synthesis of stilbenes was observed during the last stage in both processes, as well as higher accumulation in the C samples.

### 3.6. Accumulation of Glycoside Aroma Precursors in Berry

Glycoside aroma precursors identified in Corvina grape belong to different chemical classes of compounds, and their composition in the grape berry expressed as μg IS/kg d.g., is reported in Appendix A. Appendix A show the contents of the single compounds in the two years of study. Total amounts of each class of glycoside aroma precursors at the three withering stages are reported in Table 1.

Glycoside aliphatic alcohols identified were 1-butanol, 3-methyl-1-butanol, 3-methyl-2-butene-1-ol, 3-methyl-3-butene-1-ol, 1-hexanol, 3-hexen-1-ol *E* and *Z*, 2-butoxyethanol, and 2-hexenol. In 2016, C samples had higher levels of total aliphatic alcohols, but a significant difference (*p* ≤ 0.05) between C and NC samples was found only at the highest content at −20% WL stage. In 2017, the total content was higher in NC with a significant difference at −20% and −30% WL, and the highest aliphatic alcohols content was reached at −30% WL in both C and NC (Table 1).

Glycoside derivatives of C_6_-aldehydes hexanal and 2-hexenal, were identified; the aglycones confer to the wines herbaceous/grassy notes. Their content was higher in NC in both years, but, a significant difference between C and NC was found at −10% and −30% WL in 2016 and at −20/−30% WL in 2017. In both years, the highest concentration was achieved at −20% WL.

Fourteen glycoside monoterpenes were identified in Corvina grape, which include furanlinalool oxide *cis* and *trans*, *trans*-pyranlinalool oxide, linalool, α-terpineol, nerol, geraniol, diendiol I, 8-hydroxylinalool *cis* and *trans*, hydroxygeraniol, 2-exo-2-hydroxycineol, 7-hydroxy-α-terpineol and geranic acid. The highest levels were found for geraniol and 7-hydroxy-α-terpineol. In general, the aglycones are characterised by low sensory thresholds and confer to the wines floral or citrus notes [36]. In 2016, monoterpene total content was higher in C at all withering stages, a significant difference between C and NC was found at −10% and −20% WL. As expected, water loss increased the monoterpene concentration in the berry and the highest content was found thereabout at −30% WL in both C and NC, but no statistical difference among the two processes was found. Indeed, in 2017 the content was higher in NC at all withering stages and a significant difference between the two processes was found at −20% and −30% WL. In general, the highest concentrations in the berry were reached in the last withering stages, showing that the process does not promote the degradation processes of these glycoside precursors.

C_13_-Norisoprenoids are correlated to floral/spicy notes arising in red wines in particular after aging [36]. Glycoside norisoprenoids identified in Corvina grape are 3-hydroxy-β-damascenone, 3-oxo-α-ionol, 3,9-dihydroxy-megastigma-5-ene, 3-hydroxy-7,8-dihydro-α-ionol and vomifoliol. Potentially, they are precursors of volatile compounds which contribute by conferring positive notes to the aroma of wines, such as β-damascone (fruity note), β-damascenone and 3-oxo-α-ionone (floral, tobacco). In 2016, total content of norisoprenoids at −10% WL was higher in C, indeed, at −20% and −30% WL the two processes showed similar levels of compounds. In 2017, the content was higher in NC samples at all withering stages with a significant difference between the processes at −20% and −30% WL. In this year, the highest concentration of norisoprenoids was found at −30% WL in NC, instead in C at −20% WL.

Glycoside precursors of sixteen compounds belonging to the chemical class of benzenoid derivatives were identified: benzaldehyde (almond note), acetophenone, methyl salicylate, guaiacol, benzyl alcohol, β-phenylethanol (rose), eugenol (clove), 4-vinylguaiacol, 4-vinylphenol, syringol, vanillin (vanilla), methyl vanillate, acetovanillone, 4-hydroxybenzeneethanol, vanillic and homovanillic alcohols [36]. In 2016, their total content during the entire process was higher in NC, and a significant difference between the two processes was found at −30% WL. The highest concentration was reached at −30% WL in NC and at −20% WL in C samples. The second year of the study confirmed the higher contents in NC during withering, with a significant difference at −20% and −30% WL. In both processes, the highest accumulation of benzenoids was reached at −30% WL.

Moreover, regarding the total content of aromatic compounds at every stage, in 2016, at −10% and −20% WL, the content was almost the same in both C and NC, while at −30% WL, the content was higher in NC than in C, with a statistical difference between treatments. In 2017, there were more aromatic substances at every stage in NC than in C. Furthermore, at −20% and −30% WL, statistical differences were found between treatments, with the already-reported higher content in NC.

### 3.7. Sensory Analysis of the Resulting Wines

Results of the tastings carried out in 2016 and 2017 are reported in Figure 10. The 2016 vintage showed a great difference between the two wines. NC generally resulted as richer and more interesting regarding the olfactory scents, with low herbaceous notes and higher scores for smoothness. In the mouth, the NC wine was smoother and better appreciated in terms of finesse, taste balance, and full body. The resulting C wine achieved a lower score compared to NC, which displayed more character (see full-bodied, structure) and elegance (see finesse and olfactory quality less herbaceous).

As for the 2017 vintage, the NC wines confirmed the higher pleasantness with a peak in ripe fruit combined with spicy notes. Overall, the NC wine was more harmonious with good scores in terms of taste balance and finesse.

## 4. Discussion

The microclimatic analysis of the chambers used for tests confirmed a different air thermal and humid regime in the two situations: lower temperatures and higher humidity in NC, higher temperatures, and lower humidity in C (Figure 1 and Figure 2). Furthermore, NC temperature was consistently closest to the external thermal conditions simulating the external regime. T and RH are directly related to the kinetics of withering; they entail the transference of energy and water from the berries into the environment. Concerning this parameter, we found that the trend of weight loss was faster in C conditions than in NC, and consequently, −30% WL was reached earlier in C (2 weeks). This could have resulted because the highest T and lowest RH for C led to a faster transference of energy and water from the berries into the environment due to the higher value of vapour pressure deficit (VPD), resulting in a faster WL. Thus, we can state how the higher ventilation in C can facilitate water loss from berries due to removing the boundary layer [47,48], reducing the RH around the berry and accelerating the WL.

Plant phenolics have important effects on food quality and human nutrition. Their presence in grapes and red wine may contribute to health benefits because of their antioxidant and anticarcinogenic activities. Hence, the biosynthesis of antioxidant molecules such as anthocyanins and stilbenes represent a defence mechanism for the plant cells, used as strategy for reducing the oxidative cells damage resulting from the drought stress. Moreover, these molecules have a beneficial impact on humans that consume them through drinking the wine, because they have similar antioxidant roles in human cells. For this, the interest in the physiological role of the bioactive compounds present in plants products has increased dramatically over the last decade [15,19,49]. Finally, as it was studied, we report how many factors determine the accumulation of pigments and antioxidant elements: the level of ripeness of the grapes [50], thermal stress [51,52,53], light [54,55] and water [56]. However, the drying process affects the anthocyanins enrichment by concentration and neoformation (Figure 7 and Figure 8) [26,57].

Regarding the berry chemical composition (Figure 4), at the end of the studied period, the sugar content was higher in NC than in C. This can be related to the different climatic conditions in NC than in C (T and RH), as well as a longer dehydration period. Indeed, when observing the rate of sugar increase between the two postharvest maintenances of the grapes, the rate of increase in sugar cannot be explained only by the rate of WL, and indeed, for the same WL, there were different values in sugar enrichment between treatments. An explanation can be found in the consequences that different combinations of the climatic elements (T, RH, Wind) can produce during the dehydration process, in terms of acceleration in sugar release from the cell wall, [50] mainly galactose [51].

Measurements taken included acidity (Figure 5 and Figure 6) and its component in tartaric and malic acid. The accumulation trend was similar between treatments along the withering period, with a predominance, at the end, of total acidity in NC, mostly due to tartaric acid concentration. In more detail, tartaric acid was more stable and tended to remain unaltered and even accumulate during the withering process at the different stages of WL, evidencing, as already stated, a greater concentration in NC conditions. Malic acid performed differently, showing a smaller conservation capability due to its rapid consumption during the initial stage of grape dehydration, as previously reported [58]. Nevertheless, its enrichment throughout the drying period was almost immeasurable or very minimal. These results suggest that the most significant accumulation of sugars and acidity occurs at −30%. Hence, the faster withering in C likely led to less accumulation of these compounds due to the respiration of tartaric and malic acids as acidity strongly depending on water and temperature stress [33].

Drying process induces specific chemical changes inside the berry, and most of them involve secondary metabolites (anthocyanins and general pigments, VOCs, etc.). Previously observed drought-induced compositional changes to the grapes were transferred to the wines, with an increase in polyphenols and VOCs. Moreover, in the berry, these changes are known to take place alongside transcriptome-wide reprogramming of metabolism pathways, involving all secondary metabolites [59,60,61]. Furthermore, the drying process damages the cellular structure of the grape skin, which facilitates the ex-traction of anthocyanins [62,63]. For these instances, we performed a focused analysis of the secondary metabolism.

The accumulation of polyphenols and anthocyanins evidenced an interesting trend showing an initial positive step linked to a new polyphenol and anthocyanin synthesis [61] followed by a more stable phase (up to −20% WL). Finally, a third phase took place with the accumulation of the pigment compounds due to the concentration process. Specifically, NC tended to have more of both components than C during the withering process and, at −30% WL, significant differences were found in favour of NC. Yet, in this case, NC confirmed a higher accumulation capability, probably due to a slower and less stressful withering process linked to the higher temperature in C, capable of decreasing the synthesis or increasing the degradation of polyphenols and anthocyanins as confirmed previously by Shaked [64].

Regarding stilbenes, several studies have demonstrated that osmotic stress occurring during withering, which is induced by water loss and the higher temperatures of the chamber (30 °C), stimulates stilbenes biosynthesis [15,18,19,20,21,65,66,67]. Therefore, a higher stilbene level is considered positive as it increases the health benefits of wine. In our research, the accumulation of stilbenes in the berry (Figure 9) up to −20% WL was higher in the NC samples compared to the C ones, but in the latter case, the biosynthesis of stilbenes accelerated suddenly in the final withering stage (between 20–30% WL) by increasing their level in the berry. Reasonably, the higher air humidity and lower temperature occurring in the NC process induces a longer time to reach a −30% WL in the berry (Figure 3) and most likely with a lower cell level stress that induced a lower synthesis of stilbenes compounds.

In the glycoside aroma precursors profile, aliphatic alcohols that were found to be increased by withering in both years were butanol, 3-methyl-1-butanol, hexanol and 2-hexenol (Appendix A). In general, these compounds account for herbaceous and unripe fruit aromas, and the wine tasting can reveal their presence [22]. cis-8-Hydroxy-linalool, geranic acid and 7-hydroxy-α-terpineol were the main monoterpenes that increased at −30% WL in both withering. This class of aroma compounds is characteristic of Amarone wines, with geraniol as the main compound [68]. Significant is the presence of α-terpineol in Corvina grapes (floral note); however, the slight increase observed at the end of withering can be linked to the chemical mechanisms of rearrangement of other monoterpenols occurring during the process [69,70].

C13-Norisoprenoids contribute to the aromas of the Amarone with notes of ripe fruit, honey, jam, tea and tobacco. Here, the glycoside derivatives of 3-hydroxy-β-damascenone, 3-oxo-α-ionol and vomifoliol had the highest concentrations in NC.

The primary glycoside benzenoids (spicy, balsamic aromas) found are methyl salicylate, guaiacol, benzyl alcohol, β-phenylethanol, syringol, vanillin and vanillic alcohol. In both the dehydration processes, the highest amount of benzenoids in the berry was found at −30% WL, particularly benzyl alcohol, guaiacol, syringol, vanillin, acetovanillone and vanillic alcohol. Specifically, NC treatment evidenced the highest quantity in benzenoids due to the high level in benzyl alcohol, guaiacol, syringol, vanillin, acetovanillone, eugenol and vanillic alcohol (Appendix A) and together can contribute to the typical aroma of Amarone wines. Generally, the increase in some glycoside compounds observed during withering is linked to the concentration effect into the berry due to water loss. Moreover, by increasing dehydration temperature, the level of monoterpenes and glycoside compounds in the berry tends to decrease [26].

The general trend of the content in 4-vinylguaiacol at −30% WL (spicy note; sensory threshold in wine 40 μg/L [71] was statistically different between the two processes (*p* ≤ 0.05 and *p* ≤ 0.001 in 2016 and 2017, respectively) with higher contents in NC samples (data not shown). The low benzaldehyde level found in the samples compared to benzyl alcohol (between 20–50 µg/kg d.g. and 800–2500 µg/kg d.g., respectively) indicates that low or no oxidative processes occurred due to Botrytis cinerea infection [72].

Lastly, concerning the sensory evaluations, the wines obtained confirmed the differences found in all analyses conducted on grapes. For NC, there were greater notes of spice (due to higher levels of benzenoids where the acetovanillone, vanillic alcohol and benzyl alcohol stand out) and a greater structure (given by the whole composition), which is also accompanied by greater finesse, elegance, and olfactory richness. Overall, it was a more harmonious, structured and smooth wine (less tannicity) than C, where the unpleasant herbaceous notes tended to be higher. Similar outcomes confirming our research are reported in the literature [73,74,75].

## 5. Conclusions

In general, the drying process carried out in NC conditions was slower than in C due to higher RH, lower T and no air-forced movement in the NC environment. This slower process led to a major accumulation of sugars, acids, polyphenols and anthocyanins in NC. It is assumed that the slower process in NC was able to maintain vital cell structures for longer, with less water stress effects (lesser synthesis of stilbenes can confirm this observation). In terms of total aroma compounds, we found a higher content in NC, also confirmed by the better wine profile of these wines. Finally, to reach a considerable content of compounds and obtain balanced wines, we can report that it is desirable to reach a −30% WL and adopt NC conditions. Lastly, the sensory evaluation of wines highlighted the positive sensory descriptors present in the wines produced adopting the non-controlled process.

The present research confirmed that slowing down the drying process and creating less stressful conditions for the cells could be a strategic option to achieve a more traditional and well-appreciated Amarone wine and that choosing the traditional drying process carried out in natural conditions could also improve overall market appreciation.

## Figures and Tables

**Figure 1 molecules-26-05198-f001:**
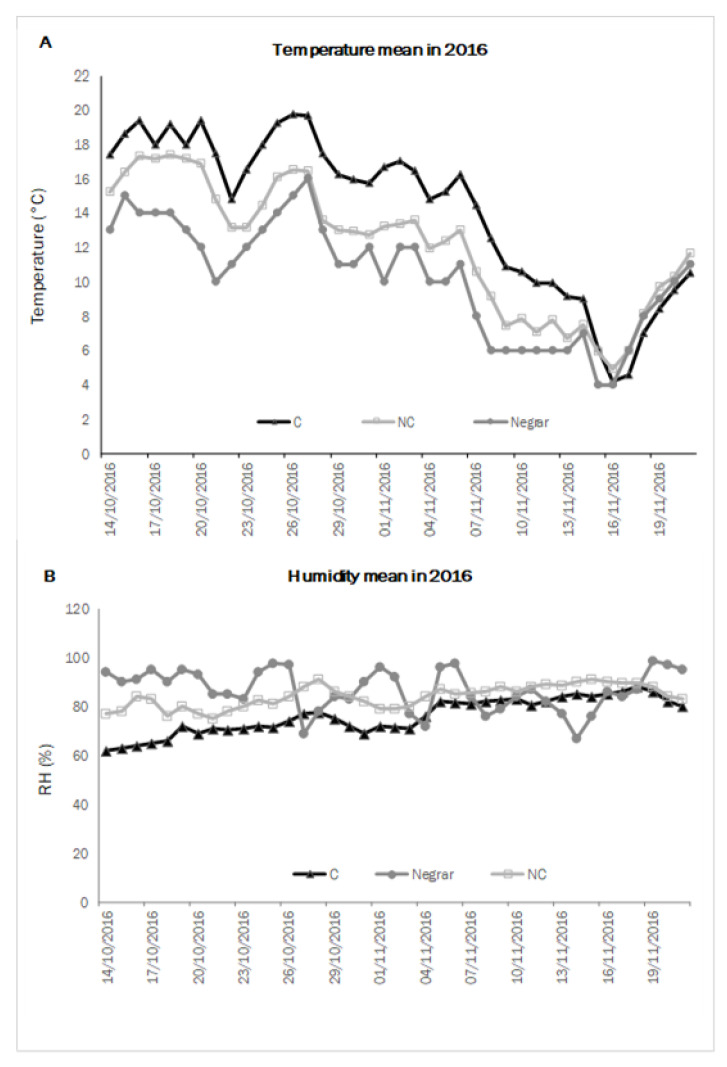
Daily mean air temperature (**A**) and humidity (**B**) in controlled (C) and not-controlled (NC) chambers and outside environment (Negrar) in 2016.

**Figure 2 molecules-26-05198-f002:**
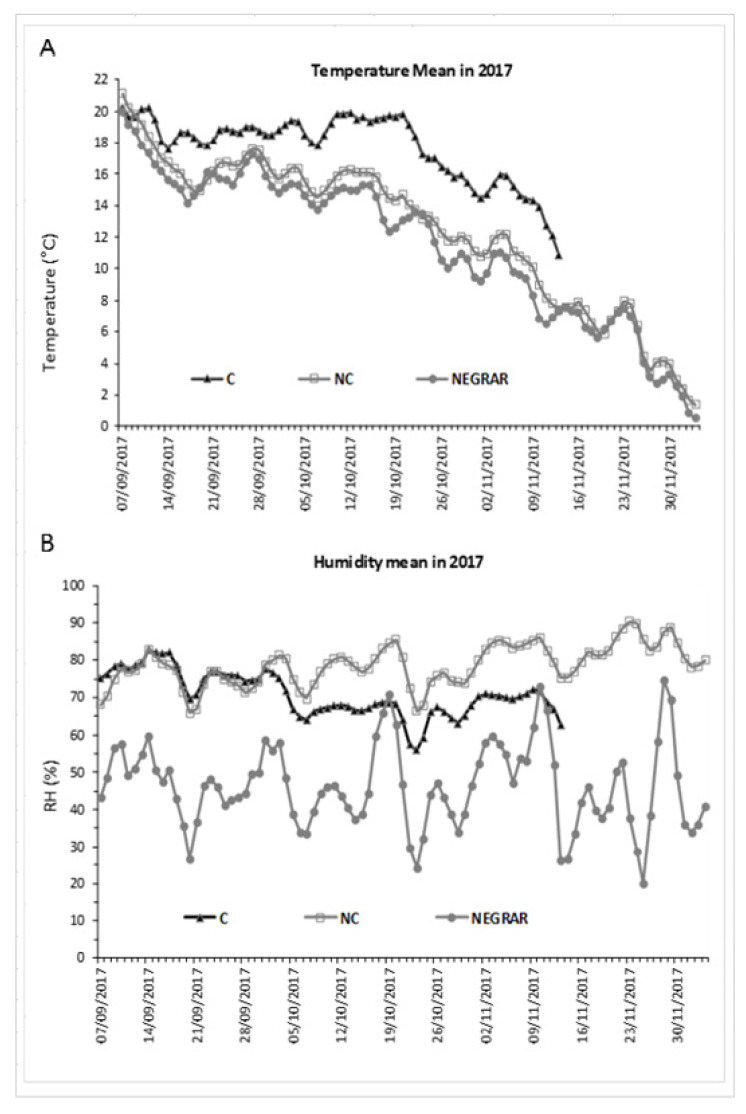
Daily mean air temperature (**A**) and humidity (**B**) in controlled (C) and not-controlled (NC) chambers and outside environment (Negrar) in 2017.

**Figure 3 molecules-26-05198-f003:**
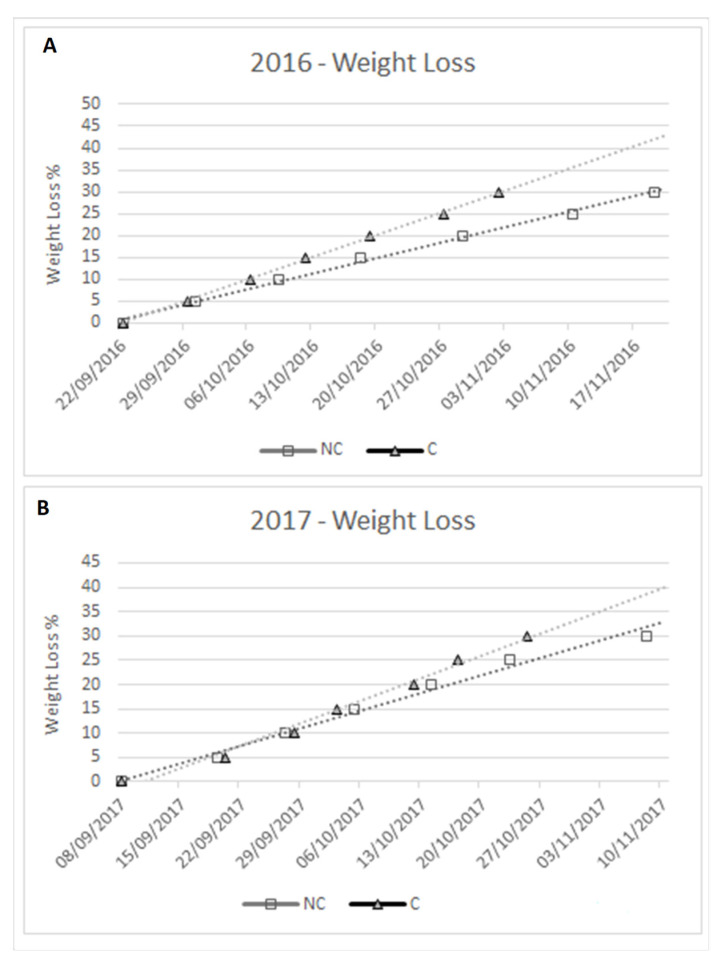
Kinetics of withering: WL in 2016 (**A**) and 2017 (**B**).

**Figure 4 molecules-26-05198-f004:**
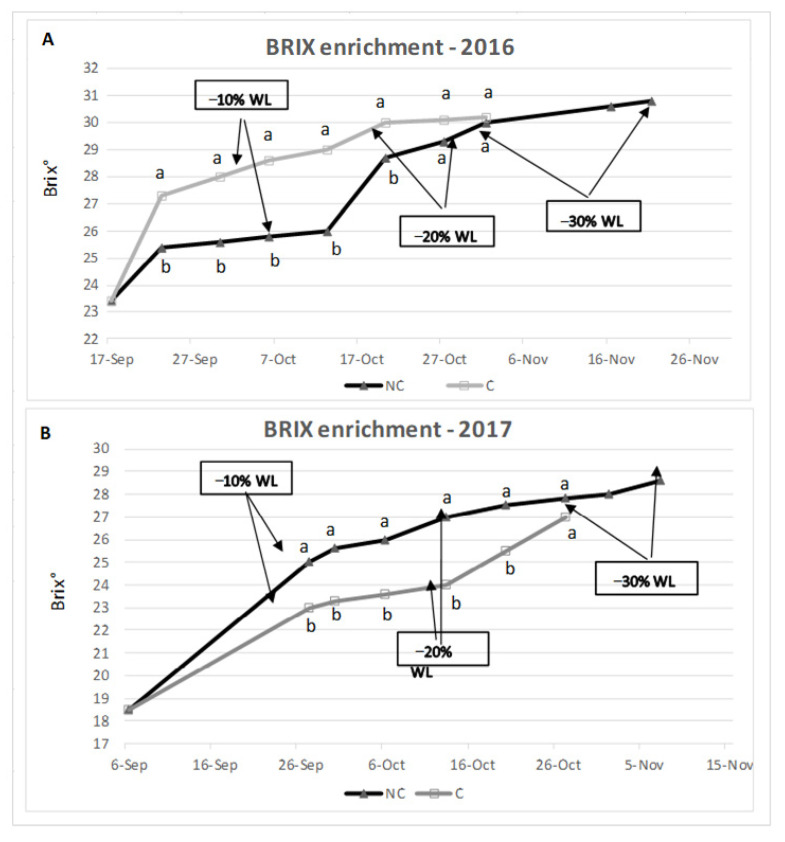
Brix degree enrichment in 2016 (**A**) and 2017 (**B**). Arrows mean different WL values. Where present, values indicated with different letters were significantly different (*p* < 0.05) between three analytical replicates.

**Figure 5 molecules-26-05198-f005:**
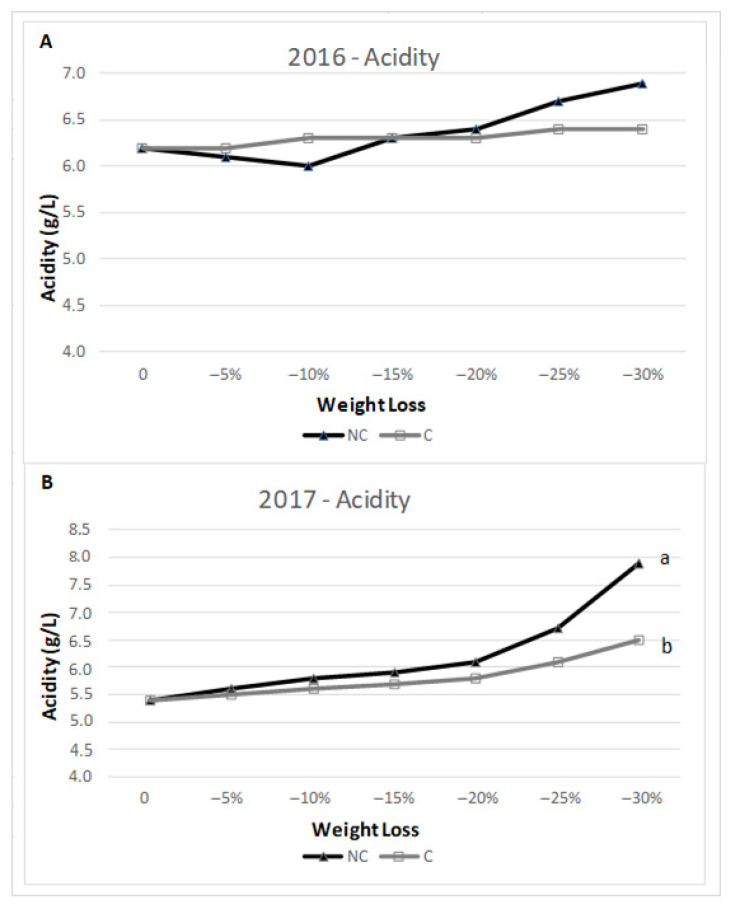
Total acidity (g/L of tartaric acid) in 2016 (**A**) and 2017 (**B**) at harvest date and different WL stages. Where present, values indicated with different letters were significantly different (*p* < 0.05) between 3 analytical replicates.

**Figure 6 molecules-26-05198-f006:**
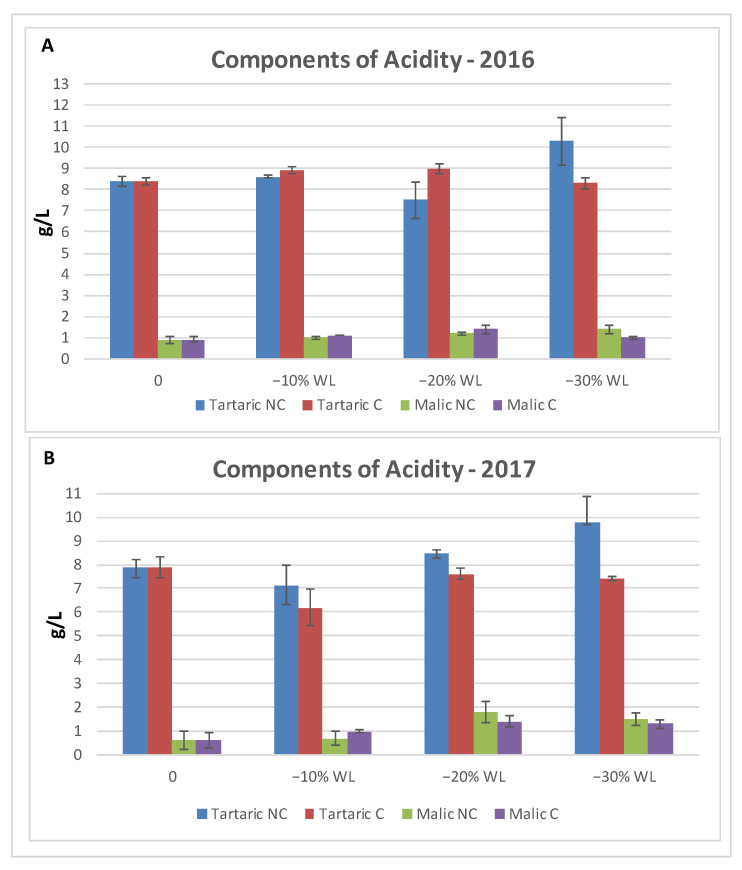
Components of total acidity in 2016 (**A**) and 2017 (**B**). Vertical bars indicate Standard Deviation of 3 analytical replicates. No statistical differences were found.

**Figure 7 molecules-26-05198-f007:**
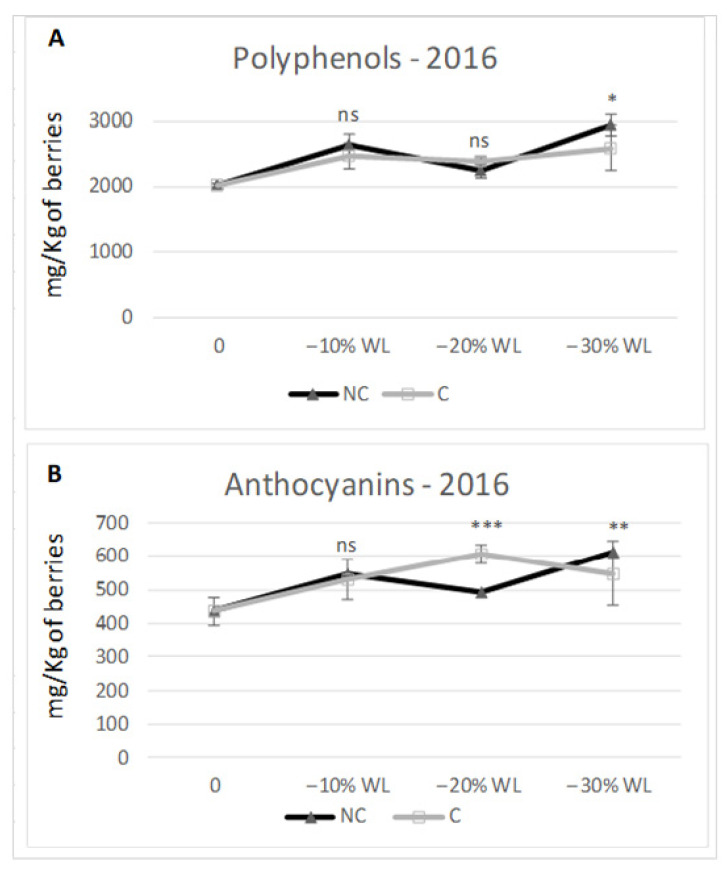
Content in polyphenols (**A**) and anthocyanins (**B**) in 2016. Data are the mean of three berry samples. *; **; *** indicate a significant difference (*p* ≤ 0.05, 0.01, 0.001) between C and NC treatment. Vertical bars indicate standard deviation of 3 analytical replicates.

**Figure 8 molecules-26-05198-f008:**
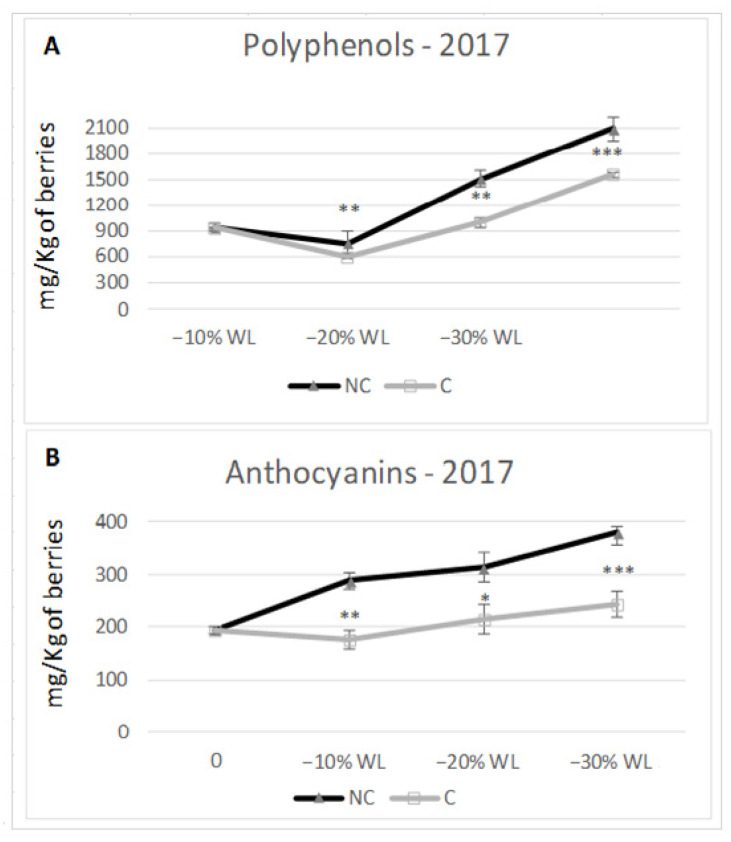
Content in polyphenols (**A**) and anthocyanins (**B**) in 2017. *; **; *** indicate a significant difference (*p* ≤ 0.05, 0.01, 0.001) between C and NC treatment. Vertical bars indicate standard deviation of 3 analytical replicates.

**Figure 9 molecules-26-05198-f009:**
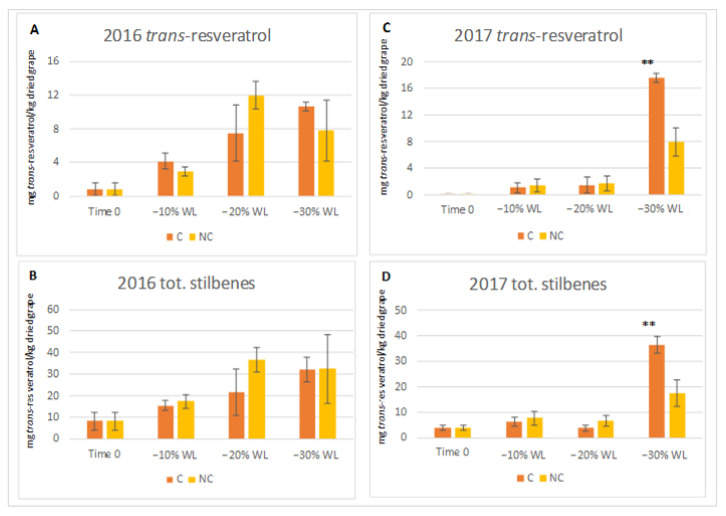
Levels of *trans*-resveratrol and total stilbenes ((**A**,**B**) for 2016, (**C**,**D**) for 2017 vintages) calculated as mg *trans*-resveratrol/kg dried grape (d.g.) during two withering processes carried out in the two years. C, dehydration in RH and WS controlled warehouse; NC, dehydration carried out in the not-controlled environment. ** indicate a significant difference *p* ≤ 0.01 between the two processes. Vertical bars show the standard deviation of 3 analytical replicates.

**Figure 10 molecules-26-05198-f010:**
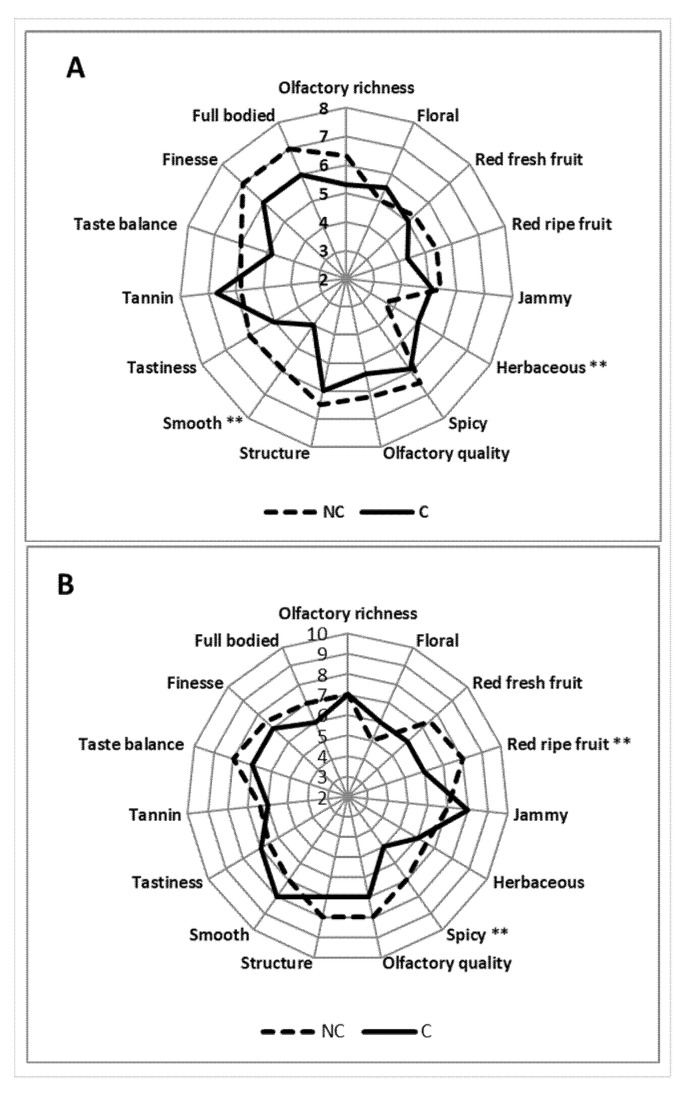
Results of tasting for wines in 2016 (**A**) and 2017 (**B**) years. ** indicate a significant difference *p* ≤ 0.01 between C and NC treatment.

**Table 1 molecules-26-05198-t001:** Glycoside aroma precursors in Corvina grape during withering.

	t_0_	−10% WL		−20% WL		−30% WL	
sum	(μg/Kg d.g.)						
2016		C	NC	C	NC	C	NC
aliphatic alcohols	299	444	335	774 **	570	575	535
C_6_-aldehydes	27	43	55 *	41	46	21	35 *
monoterpenes	553	805 *	692	862 *	877	857	813
norisoprenoids	785	1103 *	914	1149	1177	1126	1144
benzenoids	2401	4637	4728	5286	5472	4611	5805 **
**Total**	4065	7033	6724	8112	8142	7190	8332 **
**2017**							
aliphatic alcohols	485	538	460	456	552 *	641	820 *
C_6_-aldehydes	58	55	70	120	138 *	65	104 **
monoterpenes	738	625	829	640	912 **	698	895 *
norisoprenoids	1074	961	1462 *	1485	1968 **	1164	2702 **
benzenoids	3257	3258	3696	4430	6156 **	5937	8698 ***
**Total**	5612	5437	6517	7131	9726 **	7865	12,400 ***

C, in-chamber dehydration under controlled conditions of relative humidity (RH) and wind speed (WS); NC, dehydration carried out in not-controlled environment. Contents of compounds were calculated as μg internal standard (IS) 1-heptanol per kg of dried grape (d.g.). *, **, ***: significant difference between the C and NC process (*p* ≤ 0.05, 0.01, and 0.001, respectively). Singular compounds in the two years are reported in Appendix A.

## Data Availability

Not applicable.

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
