# Peer review of "Effects of Traditional and Modern Post-Harvest Withering Processes on the Composition of the Vitis v. Corvina Grape and the Sensory Profile of Amarone Wines"

_molecules, 2021, doi:10.3390/molecules26175198_

Round 1

Reviewer 1 Report

Manuscript ID: molecules-1338770

Title: Effects of Traditional and Modern Post-Harvest Withering Processes on the Composition of the Vitis v. Corvina Grape and the Sensory Profile of Amarone Wines

This paper aims to compare natural and artificial Corvina with-128 erring processes to assess their effect on dried grape chemical composition and peculiar 129 wine organoleptic evaluations. However, the manuscript is not well organized, and lacking technical merit.

Specific comments are listed as follows:

  1. Introduction Section unfocused. Line 59-64, The author points out that past studies have shown that the drying conditions influence the properties of the resulting grapes and wines; slow drying at a low temperature (T) and aeration rate (W) and higher relative humidity levels (RH), provides more harmonious wines (e.g., equilibrated, balanced ratio of every aromatic and chemical compound) [3]. Why does the author need to compare traditional and modern methods?
  2. The introduction should be clarified to this research aim. After reading the introduction, I cannot understand why the author wants to do this research and the real purpose.
  3. Tasting only invites 10 specialists, will the number of samples be too small?
  4. Line 146, The author mentioned that in modern warehouses, air dehumidifiers maintained the air humidity between 60 and 70%, but from Figure 1 and Figure 2, the humidity range of the NC group is not between 60 and 70%, and the ups and downs in Figure 2 are large. How to prove the accuracy of the experiment?
  5. The figure and table format, unfortunately. All results charts do not show statistical results or labels. It is not clear whether there is a significant difference between the data. Suggest authors afresh the figure and table, and label the results of the statistics. The format of the line chart is different. For example, the marking symbols of each group are different, the icons are also reversed. Some have X or Y axis no unit, some have titles, some are untitled, some titles are written with the year, and some are not written with the year. Need to be uniformity.
  6. Line 658-679, the title is the conclusion, but the content (1) why are there other references? (2) Does not correspond to the purpose. This section cannot match the title.

Abstract

Abstract Section is lacks logic. Unable to understand the contribution of research.

The abstract does not appear (1) differences in composition and sensory evaluation between the two methods (2) what is the conclusion of the paper?

Introduction

  1. Line 44, The author mentioned that the weight loss (WL) of grape withering is around 30% when making wine, so why does the author need to study the difference of different WL?
  2. Line 54, 66, The English of the fruit peel should be peel instead of skin.

Materials and Methods

  1. Line 134, The conditions of the traditional method and the modern method should be explained clearly, including temperature, humidity, drying rate, etc., as well as the number of grape samples used and the value of n.
  2. Line 199, 227, The English of the fruit peel should be peel instead of skin.
  3. Line 271, Once a determinate alcoholic level. What is the concentration that an alcoholic needs to reach?
  4. For devices and other products, the specific brand or trade name, the manufacturer, and their location (city, state, and country) should be provided the first time the device or product is mentioned in the text.

Results

  1. The figure and table format are, unfortunately. All results charts do not show statistical results or labels. It is not clear whether there is a significant difference between the data. Suggest authors afresh the figure and table, and label the results of the statistics.
  2. Line 327, Suddenly H appears. I don't know what it means because it hasn't been mentioned before.
  3. The result is confusion. Line 415-424, It is mentioned that the compound with the highest signal in Table S1 analysis is quantified (trans-resveratrol and total stilbenes). The quantitative result is shown in Figure 9, but the compound with the highest signal in Table S1 is not trans-resveratrol and total stilbenes.

Discussion

In the discussion section, individual indicators are describing, but lack integration discussion.

1.Line 606-609, please confirm which table of supplementary materials is correct.

2.Why is it that only 30% weight loss is used for winemaking in the end, and why are fresh, 10% weight loss, and 20% weight loss not used together for winemaking?

  1. Line 612-615, The results in Figure 9 show different trends in 2016 and 2017. Why? What are the possible reasons?
  2. Line 616-618, What are the possible reasons for the increase in the content of certain compounds mentioned in Table S3?
  3. Line 626-630, What are the possible reasons why the 2017 results are higher than the 2016 results?
  4. Line 626-630, In the 2017 results, the content of compounds with C at -30 WL is lower than -20 WL. What are the possible reasons?
  5. Line 645-657, Are there any relevant references to support this discussion?

8.There is no discussion related to the evaluation.

  1. Line 658-679, the title is the conclusion, but the content (1) why are there other references? (2) Does not correspond to the purpose.

10.Line 678-679, The author mentioned that the research results are similar to Slaghenaufi et al., 2020. Then why does the author carry out this research, and what is the difference with Slaghenaufi et al., 2020?

References

The format needs to be reconfirmed and adjusted according to the journal regulations, such as unifying the capitalization of words, italics for journals, abbreviations for journals, italics for volume, etc.

Author Response

This paper aims to compare natural and artificial Corvina with-128 erring processes to assess their effect on dried grape chemical composition and peculiar 129 wine organoleptic evaluations. However, the manuscript is not well organized, and lacking technical merit.

Corresponding author: Mian Giovanni, PhD, University of Udine.

First of all, many thanks for all your useful comments. I will respond both here and in the main text where necessary.

Specific comments are listed as follows:

  1. Introduction Section unfocused. Line 59-64, The author points out that past studies have shown that the drying conditions influence the properties of the resulting grapes and wines; slow drying at a low temperature (T) and aeration rate (W) and higher relative humidity levels (RH), provides more harmonious wines (e.g., equilibrated, balanced ratio of every aromatic and chemical compound) [3]. Why does the author need to compare traditional and modern methods? Specifically, the fact of choosing one or another withering system is an instance struggling winegrowers, due to the fact that for most of them a fast drying (artificial methods) may be important in order to speed up each winery operation and have less botrytis infections problems. This is why nowadays few wineries adopt natural drying process, but meanwhile, they have to keep in mind what might led a faster withering, as we report in our study. Anyway, I tried to explain better in the text

  1. The introduction should be clarified to this research aim. After reading the introduction, I cannot understand why the author wants to do this research and the real purpose. Thanks, I added at the end of introduction
  2. Tasting only invites 10 specialists, will the number of samples be too small? Thanks, the number may seem too small, but the panel was made by very specialists, well trained on the sensory profiles of Amarone and with and high score in accordance. So, perhaps, in some cases, it is better having less specialists but focused and with lot of acknowledgements than lot of them without any experience

  1. Line 146, The author mentioned that in modern warehouses, air dehumidifiers maintained the air humidity between 60 and 70%, but from Figure 1 and Figure 2, the humidity range of the NC group is not between 60 and 70%, and the ups and downs in Figure 2 are large. How to prove the accuracy of the experiment? Yes, many thanks. The mistake was before. The range is not 60 – 70 but 60 -80 %. In figure 2 there are larger ups and downs since the external RH and T fluctuated a lot during the experiment in 2017, yet more than in 2016

  1. The figure and table format, unfortunately. All results charts do not show statistical results or labels. It is not clear whether there is a significant difference between the data. Suggest authors afresh the figure and table and label the results of the statistics. The format of the line chart is different. For example, the marking symbols of each group are different, the icons are also reversed. Some have X or Y axis no unit, some have titles, some are untitled, some titles are written with the year, and some are not written with the year. Need to be uniformity. Many thanks. I tried to uniformise the best I could. Hence, figure 1 and figure 2 have the similar format and icons now, no statistics is present since they only are wheatear data recorded. Figure 3 is now uniformed with Fig 1 and 2. Also here, there is no statistics made for this, because, as it has been explained in the text, the weight loss was recorded for the whole trays (whole arella and entire pallet of plastic boxes). As for Fig 4 and 5, we report your suggestions. Concerning Fig. 6, I personally think no modifications are required but, however, let me know. Figures 7 and 8 have the statistics and were modified in accordance with the other figures. For what concern fig. 9, also here I personally think no changes are required, along with fig. 10 that is completely different from the others showing other data. Hopefully it might be good as I did in accordance with your kind comments and now are uniform.

  1. Line 658-679, the title is the conclusion, but the content (1) why are there other references? (2) Does not correspond to the purpose. This section cannot match the title. Here we put other references to strengthen the conclusions. Moreover, I read again the conclusion section, it seems to be focused with the title of our manuscript. I really don t know how to change it. Perhaps, the referee may indicate to me how to improve this section. Thanks

Abstract

Abstract Section is lacks logic. Unable to understand the contribution of research.

The abstract does not appear (1) differences in composition and sensory evaluation between the two methods (2) what is the conclusion of the paper? I added 2 sentences concerning the wine tasting along with a conclusion

Introduction

  1. Line 44, The author mentioned that the weight loss (WL) of grape withering is around 30% when making wine, so why does the author need to study the difference of different WL? We decided to study the drying process step by step as in other scientific publications based on the same subject.
  2. Line 54, 66, The English of the fruit peel should be peel instead of skin. In different published papers it is reported skin instead of peel. Perhaps, peel is more accurate for fruit trees. We would like to keep this format.

Materials and Methods

  1. Line 134, The conditions of the traditional method and the modern method should be explained clearly, including temperature, humidity, drying rate, etc., as well as the number of grape samples used and the value of n.

We tried to be clearer explaining more in details the two drying conditions, as for the grape samples we explained them in 2.3.

  1. Line 199, 227, The English of the fruit peel should be peel instead of skin. . In different published papers it is reported skin instead of peel. Perhaps, peel is more accurate for fruit trees. We would like to keep this format.
  1. Line 271, Once a determinate alcoholic level. What is the concentration that an alcoholic needs to reach?  Ok we add the alcohol level
  2. For devices and other products, the specific brand or trade name, the manufacturer, and their location (city, state, and country) should be provided the first time the device or product is mentioned in the text. Added where missing text

Results

  1. The figure and table format are, unfortunately. All results charts do not show statistical results or labels. It is not clear whether there is a significant difference between the data. Suggest authors afresh the figure and table, and label the results of the statistics. As the referee commented above, I uniformised each graph
  2. Line 327, Suddenly H appears. I don't know what it means because it hasn't been mentioned before. Sorry, it was RH. Corrected
  3. The result is confusion. Line 415-424, It is mentioned that the compound with the highest signal in Table S1 analysis is quantified (trans-resveratrol and total stilbenes). The quantitative result is shown in Figure 9, but the compound with the highest signal in Table S1 is not trans-resveratrol and total stilbenes. the sentence has been changed in the revised ms.

Discussion

In the discussion section, individual indicators are describing, but lack integration discussion.

1.Line 606-609, please confirm which table of supplementary materials is correct. citation have been corrected as Figure 9

2.Why is it that only 30% weight loss is used for winemaking in the end, and why are fresh, 10% weight loss, and 20% weight loss not used together for winemaking? This is because the law (wine production disciplinary) permit to obtain Amarone wine only using grape with at least 30 % weight Loss

  1. Line 612-615, The results in Figure 9 show different trends in 2016 and 2017. Why? What are the possible reasons? in the revised ms it was introduced: ”Reasonably, these differences are mainly correlated to different temperature and humidity conditions recorded in the two years during grape withering (Figures 1 and 2)”
  2. Line 616-618, What are the possible reasons for the increase in the content of certain compounds mentioned in Table S3? in the revised ms it was introduced: ”In general, the increase of some glycoside compounds observed during withering are linked to concentration effect into the berry due to water loss. Moreover, by increasing de-hydration temperature the level of monoterpenes and glycoside compounds in the berry tends to decrease [26]. In our study the temperatures used in the controlled process ( C ) were higher in both years with respect to the NC samples (Figures 1 and 2).” But we have to add too  that alongside the drying process transcriptome-wide reprogramming of metabolism pathways, involving all secondary metabolites will took place and this can partially explain the increase in some chemical compounds.
  3. Line 626-630, What are the possible reasons why the 2017 results are higher than the 2016 results? see answer 3 reported here, also considering that the 2017 temp. was higher 
  4. Line 626-630, In the 2017 results, the content of compounds with C at -30 WL is lower than -20 WL. What are the possible reasons? This is an interesting point, other aromatic compounds have the same behaviour too (ie benzenoids in C 2016) We will investigate these aspects in a new oncoming research
  5. Line 645-657, Are there any relevant references to support this discussion? No, these are only our deductions based on the results of our research. Only a similar reference was found and put, as also indicated, by the other referee

8.There is no discussion related to the evaluation. Added, thanks

  1. Line 658-679, the title is the conclusion, but the content (1) why are there other references? (2) Does not correspond to the purpose. Here we put other references to strengthen the conclusions, I also think the conclusions match the purposes of the work. Please, let me know if it is ok now

10.Line 678-679, The author mentioned that the research results are similar to Slaghenaufi et al., 2020. Then why does the author carry out this research, and what is the difference with Slaghenaufi et al., 2020? For this, we reported in the test that our results are in line with Slaghenaufi et al., 2020, because in that paperwork they investigated on wines and, as results, the wine tasting showed similar result. There is a great differences, however, among the two research, since we work primarily on berries aroma compounds at different weight loss stages, while other colleagues (Slaghenaufi et al., 2020) considered aroma compounds in wine

References

The format needs to be reconfirmed and adjusted according to the journal regulations, such as unifying the capitalization of words, italics for journals, abbreviations for journals, italics for volume, etc Thank you, done

Many thanks for the revision, hopefully we have been clear in the responses

Reviewer 2 Report

Dear Authors,

Please find the list of my comments and suggestions in the attached pdf file.

Kind regards,

Reviewer

Author Response

POINT BY POINT COMMENTS TO THE AUTHORS
Dear Authors, I have reviewed your paper titled "Effects of traditional and modern post-harvest withering processes on the composition of the Vitis v. Corvina grape and the sensory profile of Amarone wines" that you submitted for publication in the journal Molecules. I am impressed with the amount of work put in your research and I find that the topic is important and the results are interesting. However, I believe that considerable changes to your paper still need to be done before it can be considered for publication. Please find the list of my comments and suggestions below.

Many thanks to the referee by all authors. Thanks for all of yours useful comments.

Giovanni Mian, corresponding author

  • English language – although sentence structure and the use of grammar are technically correct, the text is sometimes hard to read because unusual formulations and strange constructions make it hard for the reader to focus. Examples are numerous and I will point out to you just a couple of illustrative ones below. Please have the English language of your paper thoroughly revised by a person familiar with the scientific topic of your work. Thanks, we will use the English editing of this Journal after the revisions or a contribute of a native English speaker
    o first of all – please replace "colour substances" and "colouring substances" with "pigments" everywhere in the manuscript text Done, thanks
    o line 41-42: "on the market where selling" yes
    o line 64-65: "drying alters the overall structure of the berry due to the loss of moisture and
    consequent berry sanitary issues" changed
    o line 90: "an oxidative process of the polyphenols" ("polyphenol oxidation" is meant) yes, it is clear, I think also in the text
    o line 279: "after a decanting of it 3 days later" changed
    o line 320: "the external humidity level presented more abrupt variability" changed
    o line 361: "the analysis of the two tested years reported an almost specular behaviour" done
    o line 429: "reaching the other process" (I believe that "reaching values comparable to those in the
    other treatment" was meant) yes. It is
    o line 440: "the sums of their contents" ("their total content" or "their composition" is meant) yes. That is
    o line 452: "a significant difference among C and NC thesis" (I believe that "a significant difference
    between C and NC treatments" was meant) yes, thanks
    o Several sentences are so strangely formulated that it is very hard to even guess their meaning (e.g.,
    lines 75-79, 81-82, 354-356). Thanks to point out, we will reformulate by the help of a native English speaker or by using the English editing service
    • Abstract – the Abstract is overall well written but needs some particular corrections:
    o line 21: please correct "to have the chemical metabolites concentrate in the berry" into "to
    concentrate the metabolites in the berry". All metabolites are always chemical. done
    o line 22-23: The particular local regulation is not relevant for the Abstract. You may mention it in the
    Introduction, however please keep in mind that the global readership of your paper will not care
    about particular local regulation. Instead, you may mention that in the current wine production
    practice in Valpolicella, two approaches are used, a traditional one and a modern one, and then
    describe them. Thank you done

o line 26: "open-air natural environmental conditions" is misleading, because it makes the reader think
that the traditional method is performed in the field or at least in a non-roofed facility. Please
rephrase. Done, thanks
o line 30: "non-controlled" should be written instead of "not-controlled" done
o End of Abstract: The Abstract lacks the most important information – Which wine is better? Please provide a conclusive synthesis to your results at the end of the Abstract. Done, many thanks
• Introduction – is correctly framed and well written, apart of the use of strange formulations. Ok, many thanks
• M&M – I have several minor remarks:
o photographs of withering facilities (interior of withering warehouse) and equipment (especially those
of local significance, such as arelle) would be beneficial for the paper, if available. The availability of
photos is not crucial for the manuscript, but they would be certainly helpful to the reader. Thanks for the advice, I added as supplementary material
o please provide the geographic coordinates of the location and the official data on general climatic
description (annual temperatures, precipitation, humidity etc.) ok, done, thanks
o lines 119 and 352: How do you control humidity in a NC chamber? A brief explanation should be
added to the M&M section, and also in Introduction. In that case you will not need to explain again in
Results. Added, thanks
o line 152-155: This text can as well be omitted, since it provides an additional information which will
deviate the reader's attention rather than actually help him in interpreting the results. Ok, removed
o line 155-156: Please add information on how you checked weight loss to know when the -10%, -20%
and -30% weight loss moments exactly occurred. Is the paragraph 2.3. about that? If so, please add
"described in section 2.3" to the lines 155-156. Also please add the information about how often you
recorded the weight loss data. Thanks, I did exactly you advised
o In Discussion, lines 603-604, you state: "only visually healthy grapes were collected". This information
definitely should be mentioned in the M&M section. I agree, thanks, I moved to section 2.7
o lines 171-172: 250 kg and 200 kg in total for two trays, or per tray? Please specify. I reformulated. It was 2 trays per treatment
o lines 181 and 190: "quantification", not "accumulation". Accumulation is a biological process, but
quantification is what you do as a method. Also, "Quantification of pigments" should be the title in
line 190. Yes, did it
o lines 198-199: Please rephrase to: "...enzymatic hydrolysis as previously described [34-36]". Readers
will understand the relationships between the papers when they download them. Please avoid
mentioning the names of authors within the main text, even when ref. numbers are provided next to
them. Because in a journal which uses a numbered referencing style, it is too much information and
can be confusing to the reader. Yes, I totally agree. Many thanks
o lines 226-227: Same. Please avoid using the authors names within the main text. done
o line 261: Same. done
o lines 290-291: Same. done
o line 292-293: You might decide to include the question sheet in the Supplementary Data (just a
thought; not absolutely necessary) it s a right thing to ask, I can add as Suppl. Fig. Thanks
o line 243: There is a huge gap between "10" and "L" in "10 L". Please fix that. Also please commit to
either "L" or "l" for volume designation and use it consistently throughout the manuscript. I saw only now, I think it was a problem of converting files, since some strange sign appears in the .docx. I think I fitted. Thank you
• Results, Figures and Table – I have several remarks:
o general remark: Although the publication of the paper should not depend on this, you might decide
to use RGB colours for graphs, instead of grayscale. Since Molecules is an online journal, there is no
official print version and most readers will watch the graphs on their computer screen. It is not just a
matter of aestethics – the use of colours in graphs actually reduces the reader's struggle to decipher
the data in the graph and helps him grasp the point more readily. If for the revisor it is ok, I might colour only bar charts since line charts may be ok in this way
o Figure 3: X- and Y-axes should be swapped. The timing of the experiment should be presented on the
horizontal axis because the weight loss is dependent on time, not the other way around. You can use
the same format as you did for the current version of Figure 4. Yes. Honestly, we just spoke with all authors, and we knew about this. The only matter is that we already tried several times to change as you suggested, but we got a graph that is really not understandable. Now, I tried the best I can do, and I got the graph changed in the text. Hopefully it is ok. Thanks
o Figures 4 and 5: Why are you showing sugar accumulation as a function of time, but acidity as a
function of weight loss? Is there a specific reason that you chose two different ways of
representation for these two parameters? If not, I believe that both parameters should be shown as
function of weight loss as you did in Figure 5 – because both parameters are probably more directly
related to weight loss than to the actual timescale. As response I would like to say that fig. 4 was created by time in the X axes and putting the arrows of the WL in the graph in order to show both of them and to give an idea of the time elapsed in the whole analysis period, also because Brix degree tend to be more taken into account than acids, by winegrowers. Indeed, in fig 5, we thought to only show acids in relation to WL, since an idea of the whole period is just showed in fig. 4
o Figures 5 and 6: You might consider merging them into a single figure composed of 4 parts. The
content of these figures is mutually related and is discussed together. It would be easier for readers
to follow the narrative if they are able to look at both of them together. I tried but merging them resulted in a too much big figure to see every detail. If it is ok, we would like to keep them separated
o Figure 6: "No significant differences were found". Is that between the time points, or between NC
and C? In lines 371-372 you said that tartaric acid was consistently higher in NC than in C in 2017. We
can see that in the graphs, although at -10% and -20% I would say that this difference is probably not
statistically significant. Is it significant at -30%? Please be more clear about all of it in the text. Rephrased in lines 371-372

o Figure 9: Inverse colours are used for NC and C treatments compared to the previous figures, which
appears confusing to the reader. Please stick to the same choice of colours for the same treatments
throughout the manuscript regardless of whether you opt for grayscale or for colour graphs. Many thanks, I just fixed it also according to the referee 2
o Figure 10: please correct "herbaceus" into "herbaceous" and "finess" into "finesse". Ok done

o Table 1 is too tiny, it should be organised differently so that the lettering is visible without the need
to zoom in. I believe that the MDPI journals allow the authors to change the orientation of a single
page (from portrait to landscape) in order to show data such as those presented in your Table 1. the table was changed by introducing higher-size letters and numbers
o lines 397-401: If you have found differences between total anthocyanins but not between particular
types, you might even skip mentioning the particular types. A significant difference in total
anthocyanins is a good information and sounds complete by itself. If you mention additionally the
lack of difference between individual types, you will leave the reader with more questions than
answers. in the revised ms it was specified: “No relevant differences between the two processes in the qualitative profile of anthocyanins, were observed (data not shown).”

  • Interpretation and Discussion of Results – Overall, I find that your results are unfortunately not properly discussed and synthesised. The Discussion section looks more like a repetition of the Results section, rather than its interpretation and synthesis. This is also the most serious issue with your paper. I am giving you below some ideas on how you might better discuss and synthesise your results:
    o the first paragraph of Discussion is good
    o dehydration contributes to the enrichment in various antioxidant molecules not only through
    concentration (I believe you said that at one point as well); antioxidants are actively synthesised in
    dehydrated plant tissue as part of drought-stress reponse. This is because drought stress causes
    oxidative stress in plant tissues, and synthesis of antioxidants is an adaptive mechanism. A notorious example of this process is anthocyanin synthesis. This mechanism is extremely important for the
    production of secondary metabolites in vine berries during withering, but I don't remember reading
    about it clearly anywhere in your paper. You need to bring up this process in the Discussion regarding
    anthocyanin (and also other metabolites with antioxidant properties that you measured in your
    study) and support it by citing appropriate references (ideally if it has been researched in Vitis
    vinifera
    , even better if there is data for cv. Corvina). Similarly, the enrichment in other secondary
    metabolites is a biological process and deserves particular discussion of the mechanistic link, and
    also support by appropriate references, as available. This is of particular importance for publication in
    a journal where the molecular aspect is the focus. I reformulated some points in the discussion, please see there. Many thanks
    o the claims about particular molecules giving a taste of particular aromas also need references, where
    possible (even if it is in the Results section) ref [36] which reports the odour descriptors for several compounds cited was reported in lines 458, 468 and 482 of revised ms
    o linking the output (= wine composition and quality) with the input (= the two withering approaches,
    i.e., the temperature and humidity conditions measured in each of the four withering events) is
    essential for the conclusions of your work. You have measured a multitude of parameters but the
    synthesis of your results is not obvious. This is partly overcome with presenting the results of the
    wine-tasting panel, but I still feel like your results need further synthesis. A simple addition to your
    paper which should not require too much effort from your side but can give you good material for
    synthesis, could be a thorough Pearson's correlation analysis between the parameters presented in your work. Correlation between various parameters including climatic conditions (TºC, RH), chemical
    composition of the wines (like, concentration of individual molecules at the moment of -30% WL) and
    the final output (i.e., scores for particular qualities of the wine obtained from the tasting panel) can
    provide an interesting synthesis of your results and might give you strong argument to support your
    claims and draw conclusions on the best withering approach. I tried to add explanations in order to make all of this clearer without doing other statistical analyses, since the paper is just full of data and we don t want to add something else (also a comment made by the other referee reported there are just lots of results. Many thanks
    o Relate your work more to the works that are similar to yours (refs. 7, 9, 21, 24, 47, 63, 72). Check the
    discussion in those papers for ideas. Also, apart from the papers that you are already citing in your
    manuscript, you might want to check the following four and get additional ideas: Thanks, I found better adding 2 references, Degu et al 2021 and Savoi et al., 2020
    § Toffali et al. 2011
    § Bellincontro et al. 2016
    § D'Onofrio et al. 2019
    § Degu et al. 2021
    o The fact that you performed the measurements across two harvest seasons (2016 and 2017) provides
    additional value to the results, because it gives more information than what could be concluded from
    only one harvest season. However, in my opinion this abundance of results is not sufficiently
    exploited as the particular differences between the wines obtained in two particular harvest seasons
    are not of particular scientific importance (the particular differences between the 2016 and 2017
    harvest are not scientifically relevant to anyone if they don't lead to broader conclusions). In my
    opinion, there are two possible ways in which these results could be better exploited:
    § Either provide comparison between the two growing seasons (i.e., emphasise the variations
    in particular parameters between seasons, and compare the degree of season-related
    variation of parameters (2016 vs. 2017) to the degree of treatment-related variation (NC vs.
    C) – it would be interesting to see whether the treatment-related variation is more significant
    for various parameters than the season-related one – that will provide an additional
    information about the actual relevance of the treatment-related variations in wine quality) (I
    personally think this approach would be the most useful one, although you can get criticism
    for drawing conclusions based on variation between only two harvesting seasons);
    § Or merge the data from both seasons to get more robustness for the statistical analysis.
    § In any case – please avoid emphasising differences between the 2016 and the 2017 season in
    individual parameters, especially if those differences are not consistent/not relatable to the
    climatic conditions or to the differences in wine tasting – i.e. if they do not lead to a synthetic
    conclusion. Ok, thanks. As you can see in the text, I think highlighting year differences in the result section is to be applied, hence I tried to improve the discussion section, removing sentences that might be misunderstanding (regarding vintages). I also tried to not focus only on year differences as you indicated if differences were not so big, whilst adding some sentences to remark where differences were quite greats.
    • Conclusions – The Conclusions section needs to be re-written from scratch to provide meaningful synthesis
    to your findings. As a general rule, the Conclusions section of a scientific paper should not contain reference
    citing. The part where you cite the references 70-72 is good material, but for the Discussion section and not
    for the Conclusions. Also please avoid the use of abbreviations in the Conclusions – the reader should be able
    to understand your Conclusions section without reading the rest of the manuscript. Many thanks, I corrected it
    • References – Some references are not correctly formatted, please fix that. Fixed, many thanks

Once again, thank for the revision, hopefully I have been clear in the responses

Best regards,

Giovanni Mian; PhD at University of Udine (IT)

Round 2

Reviewer 1 Report

  1. There is a serious problem with the article format. The legend is to write Figure, but the article uses Fig. (e.g. Line 336), which needs to be corrected.
  2. The references really need to be reconfirmed and unified, such as the author’s name (some names are first, such as Line 832), the capitalization of single words (such as Line 803), whether the journal should be abbreviated or not, and some journals use abbreviations with dots (for example, J. Agric. Food Chem.), but some do not (such as J Exp Bot) and so on. In addition, there are not a serial number of references (for example, Line 836), please confirm.
  3. The English of the fruit peel should be peel instead of skin. In the comments of the author mentioned: [In different published papers it is reported skin instead of peel. Perhaps, the peel is more accurate for fruit trees. We would like to keep this format.] Is it possible to submit references to support?
  4. Materials and Methods

Line 165 The humidity is controlled between 60 and 80%, why choose this range? Suggest adding references.

  1. Results
  • The author presents the Y-axis unit in Figure 5 as Acidity/g/L), please confirm whether there are too many brackets.
  • Figure 6 shows the Acidity results in 2016 and 2017, but the figures above and below are all presented as 2016. Please confirm whether there are any marking errors.
  • Figures 7 and 8 are the result of polyphenols and anthocyanins, but the figures all show 2016, the must confirm that Figures 7 and 8 are 2016 or 2017.
  1. Discussion

Line 690-700 Discussion of sensory is recommended to add references to support.

Author Response

  1. There is a serious problem with the article format. The legend is to write Figure, but the article uses Fig. (e.g. Line 336), which needs to be corrected. Done everywhere
  2. The references really need to be reconfirmed and unified, such as the author’s name (some names are first, such as Line 832), the capitalization of single words (such as Line 803), whether the journal should be abbreviated or not, and some journals use abbreviations with dots (for example, J. Agric. Food Chem.), but some do not (such as J Exp Bot) and so on. In addition, there are not a serial number of references (for example, Line 836), please confirm. Serial numbers corrected and in accordance with the whole cited references both in the text and in the references section. Indeed, the journal abbreviation, with or without dots, and each cititaion are in accordance with the linked journal reference style exported by using Mendeley and crossref, already checked. Lastly and however, I modified the author names listed as the referee indicated and make some modification in accordance with the referee advice, where possible
  3. The English of the fruit peel should be peel instead of skin. In the comments of the author mentioned: [In different published papers it is reported skin instead of peel. Perhaps, the peel is more accurate for fruit trees. We would like to keep this format.] Is it possible to submit references to support?

Yes, here they are:

Vanessa Ferreira, Fátima Fernandes, Olinda Pinto-Carnide, Patrícia Valentão, Virgílio Falco, Juan Pedro Martín, Jesús María Ortiz, Rosa Arroyo-García, Paula B. Andrade, Isaura Castro, Identification of Vitis vinifera L. grape berry skin color mutants and polyphenolic profile, Food Chemistry, Volume 194, 2016, Pages 117-127, ISSN 0308-8146, https://doi.org/10.1016/j.foodchem.2015.07.142.

Muhammad Shahab, Sergio Ruffo Roberto, Saeed Ahmed, Ronan Carlos Colombo, João Pedro Silvestre, Renata Koyama, Reginaldo Teodoro de Souza, Relationship between anthocyanins and skin color of table grapes treated with abscisic acid at different stages of berry ripening,

Scientia Horticulturae, 2020,  https://doi.org/10.1016/j.scienta.2019.108859.

Zhenchang Liang, Benhong Wu, Peige Fan, Chunxiang Yang, Wei Duan, Xianbo Zheng, Chunyan Liu, Shaohua Li, Anthocyanin composition and content in grape berry skin in Vitis germplasm,

Food Chemistry, 2008. https://doi.org/10.1016/j.foodchem.2008.04.069.

4.Materials and Methods

Line 165 The humidity is controlled between 60 and 80%, why choose this range? Suggest adding references. Added, it is reported in Amarone disciplinary

5.Results

  • The author presents the Y-axis unit in Figure 5 as Acidity/g/L), please confirm whether there are too many brackets. Thank you, corrected
  • Figure 6 shows the Acidity results in 2016 and 2017, but the figures above and below are all presented as 2016. Please confirm whether there are any marking errors. Done, sorry, it was a mistake
  • Figures 7 and 8 are the result of polyphenols and anthocyanins, but the figures all show 2016, the must confirm that Figures 7 and 8 are 2016 or 2017. Done, sorry, it was a mistake once again

6.Discussion

Line 690-700 Discussion of sensory is recommended to add references to support. Added, thank

Many thanks for the useful corrections. Best regards

Referee number 2 _ second round of revisions

Reviewer 2 Report

Dear Authors,

I have performed the second round of reviewing your manuscript "Effects of traditional and modern post-harvest withering processes on the composition of the Vitis v. Corvina grape and the sensory profile of Amarone wines" that you submitted for publication in the journal Molecules. I can see that you have addressed many of the issues that I turned your attention to in the first round of review, resulting in considerable improvements in the Abstract, Introduction, Material & Methods, Results section, and Figures. The additions to the Supplementary Data are also valuable and I believe they very successfully complement your paper. Besides, the addition to the end of Introduction (lines 135-140) where you explain the importance of your work, is excellent. As I said during the first round of review, I find the purpose of your work important, and you are presenting here some very interesting results, which have great importance and significance for the wine production sector. However, the discussion of your results remains a serious issue and needs to be considerably changed – if not re-written – to be considered for publication in a journal with IF = 4.4.

Please keep in mind that the purpose of the Discussion section is not to try to provide justification for every bizarre detail regarding your results, but rather to offer generalisation and synthesis. The Discussion should not sound like a repetition of the Results section enriched with justifications or speculative explanations for one particular value in the results being greater than another particular value. That is of interest to no one.

In your Discussion section the years 2016 and 2017 are mentioned so many times, that reading it I wondered, at various points, if the point of your work was to compare the C and NC treatments, or the differences between the 2016 and 2017 vintages. Please keep in mind that the differences between the 2016 and 2017 vintages are of no interest for your readers. If your results are important and valuable (which I assure you they are) your Discussion should offer a generalisation that reaches beyond 2016 or 2017. If you want to help the wine producers in 2021, 2022, or 2058, you have to focus on general patterns, instead of bizarre differences between 2016 and 2017.

Performing measurements in two consecutive seasons instead of just one should be a strength, not a weakness of your work. Building your discussion around the differences between 2016 and 2017 turns it into a weakness; inventing speculative explanations and new, non-quantifiable parameters such as "abrupt seasonal variability" to try to justify those differences did not help. My advice would be: re-write the Discussion completely (the first paragraph might stay) and try, if possible, not to mention 2016 or 2017 in the Discussion at all. You want your Discussion focused around the differences between C and NC treatments, not around 2016 and 2017. Emphasise the mechanistic link between the phases of the process (e.g., between cellular drought stress and the biosynthesis of sugars, acids and antioxidants) and the differences between C and NC, and support them with appropriate literature citations. Also, the length of the Discussion section is not critical as long as it gives clear synthesis and generalisation of your results. A Discussion section that is half its current length can be good enough if its points are clear and sharp.

Of course, same goes for the Conclusions section.

Other than that, although you corrected certain strange and non-scientific formulations, you left many of them unchanged in the text. I am listing some of them here below and I urge you to rephrase them to adjust the language of your article to the standards of scientific language:

  • line 72: "berry sanitary issues" – please rephrase to whatever it was supposed to mean
  • line 96: "new synthesis" – please correct to "de novo biosynthesis"
  • line 97-98: "an oxidative process of polyphenols" – please correct to "polyphenol oxidation"
  • line 365 and line 388: "reported an almost specular behaviour" – please rephrase this. Scientific language should be clear and precise, and "an almost specular behaviour" is neither clear nor precise.
  • line 409: please change subtitle to "Pigment accumulation dynamics"
  • line 453: "reaching the other process" – please correct to "reaching values comparable to those in the other treatment"
  • line 455: "a sudden increase" – please correct to "a sharp increase"
  • line 465-466: "the sums of their contents... are reported" – please correct to "their composition... is reported"

Other minor remarks:

  • line 127: fans is a legitimate English word and should not be written with quotes
  • line 149: designations "N" and "E" are missing next to the respective geolocalisation coordinates.
  • line 171: Were there some precise, rigid criteria for what "dry" and "humid" days meant? Like, is the decision on opening/closing windows left to the warehouse supervisor's internal feeling, or were there some more specific criteria? In case some more specific criteria exist, they should be listed here.
  • Figure 8: I believe that Figure 8 was accidentally replaced with a repetition of Figure 7.
  • Figure 10: Some letters are jammed with the graph ("olfactory quality") or divided in two rows ("herbaceous"). Please correct that.
  • there is a non-numbered reference between ref. 36 and 37. Please double-check the reference numbering.

Author Response

POINT BY POINT COMMENTS TO THE AUTHORS
Dear Authors,
I have performed the second round of reviewing your manuscript "Effects of traditional and modern postharvest withering processes on the composition of the Vitis v. Corvina grape and the sensory profile of Amarone wines" that you submitted for publication in the journal Molecules. I can see that you have addressed many of the issues that I turned your attention to in the first round of review, resulting in considerable improvements in the Abstract, Introduction, Material & Methods, Results section, and Figures. The additions to the Supplementary Data are also valuable and I believe they very successfully complement your paper. Besides, the addition to the end of Introduction (lines 135-140) where you explain the importance of your work, is excellent. As I said during the first round of review, I find the purpose of your work important, and you are presenting here some very interesting results, which have great importance and significance for the wine production sector. However, the discussion of your results remains a serious issue and needs to be considerably changed – if not re-written – to be considered for publication in a journal with IF = 4.4. Many thanks for the appreciation
Please keep in mind that the purpose of the Discussion section is not to try to provide justification for every bizarre detail regarding your results, but rather to offer generalisation and synthesis. The Discussion should not sound like a repetition of the Results section enriched with justifications or speculative explanations for one particular value in the results being greater than another particular value. That is of interest to no one. In your Discussion section the years 2016 and 2017 are mentioned so many times, that reading it I wondered, at various points, if the point of your work was to compare the C and NC treatments, or the differences between the 2016 and 2017 vintages. Please keep in mind that the differences between the 2016 and 2017 vintages are of no interest for your readers. If your results are important and valuable (which I assure you they are) your Discussion should offer a generalisation that reaches beyond 2016 or 2017. If you want to help the wine producers in 2021, 2022, or 2058, you have to focus on general patterns, instead of bizarre differences between 2016 and 2017.
Performing measurements in two consecutive seasons instead of just one should be a strength, not a weakness of your work. Building your discussion around the differences between 2016 and 2017 turns it into a weakness; inventing speculative explanations and new, non-quantifiable parameters such as "abrupt seasonal variability" to try to justify those differences did not help. My advice would be: re-write the Discussion completely (the first paragraph might stay) and try, if possible, not to mention 2016 or 2017 in the Discussion at all. You want your Discussion focused around the differences between C and NC treatments, not around 2016 and 2017.
Emphasise the mechanistic link between the phases of the process (e.g., between cellular drought stress and the biosynthesis of sugars, acids and antioxidants) and the differences between C and NC, and support them with appropriate literature citations. Also, the length of the Discussion section is not critical as long as it gives clear synthesis and generalisation of your results. A Discussion section that is half its current length can be good enough if its points are clear and sharp.
Of course, same goes for the Conclusions section.

Author answer of the above requested changes: ok, thanks you. I am a little bit confused. It seemed to me that in the first round the referee ask me to highlight the differences between tested years and now they are not so important. I agree now, because the trends are similar despite few differences as normally it is among different years. At this point, I tried do the better I could, to improve everything as the referee kindly advised above (erasing differences within years, shorten, emphasize, etc). As last consideration, something should remain, as the other referee indicated in the first round. Also because the other referee indicated only few modifications to the Discussion and Conclusion sections, so I do not want to completely change in order not to have a negative feedback by the other referee. Please, see also the other referee comments in case you need. At the end, I tried to make a discussion and conclusion section following both referees then piecing together all revisions. Hopefully now it is ok, thank you for the detailed comment
Other than that, although you corrected certain strange and non-scientific formulations, you left many of them unchanged in the text. I am listing some of them here below and I urge you to rephrase them to adjust the language of your article to the standards of scientific language:
• line 72: "berry sanitary issues" – please rephrase to whatever it was supposed to mean Done
• line 96: "new synthesis" – please correct to "de novo biosynthesis" Corrected
• line 97-98: "an oxidative process of polyphenols" – please correct to "polyphenol oxidation" Corrected
• line 365 and line 388: "reported an almost specular behaviour" – please rephrase this. Scientific
language should be clear and precise, and "an almost specular behaviour" is neither clear nor precise. Corrected
• line 409: please change subtitle to "Pigment accumulation dynamics" done
• line 453: "reaching the other process" – please correct to "reaching values comparable to those in the
other treatment" changed
• line 455: "a sudden increase" – please correct to "a sharp increase" corrected
• line 465-466: "the sums of their contents... are reported" – please correct to "their composition... is
reported"  done
Other minor remarks:
• line 127: fans is a legitimate English word and should not be written with quotes ok thank you
• line 149: designations "N" and "E" are missing next to the respective geolocalisation coordinates. Yes, added
• line 171: Were there some precise, rigid criteria for what "dry" and "humid" days meant? Like, is the
decision on opening/closing windows left to the warehouse supervisor's internal feeling, or were there
some more specific criteria? In case some more specific criteria exist, they should be listed here. Added a little explanation, thanks
• Figure 8: I believe that Figure 8 was accidentally replaced with a repetition of Figure 7. Corrected, many thanks
• Figure 10: Some letters are jammed with the graph ("olfactory quality") or divided in two rows
("herbaceous"). Please correct that. Yes, corrected
• there is a non-numbered reference between ref. 36 and 37. Please double-check the reference
numbering. Yes, done, many thanks

All authors thank the referee for the whole corrections made till now and hopefully we have been clear following your useful advice in order to improve our manuscript.

Best regards,

Dr. Giovanni Mian, UniUD

Round 3

Reviewer 1 Report

In general, the experimental work presented is fine and the results are clearly presented.

The author's Response is corrected in the manuscript. I think this paper merits publication in Molecules.

Author Response

Many thanks to the referee for the kind revision.

All the the best,

Giovanni Mian

Reviewer 2 Report

Dear Corresponding Author,

I have performed the third round of reviewing your manuscript "Effects of traditional and modern post-harvest withering processes on the composition of the Vitis v. Corvina grape and the sensory profile of Amarone wines" that you submitted for publication in the journal Molecules. The corrections made to your manuscript during the revision process have yielded a visible improvement, especially in the Discussion and Conclusion, and I am quite pleased with the outcome.

A couple more corrections remain to be done though, for the manuscript to become complete and ready for publication. (Please see comments in the attached pdf file).

Author Response

Thank you so much.

Round 4

Reviewer 2 Report

Dear Corresponding Author,

I am recommending your article for publication in Molecules.